# Sea Breeze-Driven Daytime Vertical Distributions of Air Pollutants and Photochemical Implications in an Island Environment

Bohai Li[1], Shanshan Wang[1,2], Zhiwen Jiang[1], Yuhao Yan[1], Sanbao Zhang[1], Ruibin Xue[1], Yuhan Shi[1], Chuanqi Gu[1], Jian Xu[3], Bin Zhou[1,2,4]

[1]Shanghai Key Laboratory of Atmospheric Particle Pollution and Prevention (LAP[3]), Department of Environmental Science and Engineering, Fudan University, Shanghai, 200438, China.
[2]Institute of Eco-Chongming (IEC), Shanghai, 202151, China.
[3]National Space Science Center, Chinese Academy of Sciences, 100190 Beijing.
[4]Institute of Atmospheric Sciences, Fudan University, Shanghai, 200438, China.

*Correspondence to:* Shanshan Wang (shanshanwang@fudan.edu.cn)

**Abstract.** Atmospheric pollutants in island and coastal environments are profoundly modulated by sea breezes (SB) and typhoons. Understanding the characteristics of pollutants and their photochemical indicators under different airflow regimes is crucial for effective pollution control. Utilizing Multi-Axis Differential Optical Absorption Spectroscopy (MAX-DOAS) and a sea-land breeze objective identification algorithm, we reveal distinct spatiotemporal patterns in $NO_2$, HCHO, CHOCHO, and associated photochemistry under varying airflow patterns in a rural coastal area of Hainan, China during the summer of 2024. Non-sea breeze days (NSBDs) showed higher pollutant levels with broader vertical distribution range under conducive meteorological conditions. Conversely, on sea breeze days (SBDs), SB limits pollutant dispersion and its cooling effect suppresses ozone formation. Furthermore, SB also enhances transport of $NO_2$ and biogenic volatile organic compounds (BVOC) below 300 m, influencing ozone formation sensitivity (OFS) throughout SB phases. Typhoons effectively scavenge pollutants via strong winds and precipitation but also facilitate mid-to-high altitude BVOC transport and vertical dispersion of surface pollutants. Photochemical indicator analysis (HCHO/$NO_2$ [FNR] and CHOCHO/$NO_2$ [GNR]) indicates that the VOC-limited regimes persist even at high altitudes during typhoons. Elevated FNR and GNR thresholds suggest that existing OFS classification thresholds are inadequate for low-$NO_2$ tropical coastal rural areas, underscoring the need for region-specific assessments. Given the BVOC-dominated environment and additional inputs from SB and typhoons, GNR proves more reliable than FNR for OFS determination. This study emphasizes the necessity of integrating local meteorology and

environment conditions in $O_3$ control strategies, providing a scientific basis for pollution mitigation in tropical coastal regions prone to SB and typhoons.

## 1 Introduction

Vertical gradients in atmospheric temperature, humidity, and solar radiation, among others, establish stratified photochemical environments, while atmospheric dynamics drive spatiotemporal heterogeneity in pollutant distribution, collectively rendering tropospheric chemistry evolution exceptionally complex (Cooke et al., 2010; Gordon, 1987; Jiang et al., 2008; Mason, 2013; Wang et al., 2024). As typical photochemical precursors, nitrogen dioxide ($NO_2$), formaldehyde (HCHO), and glyoxal (CHOCHO) exhibit unique vertical distributions in the troposphere. Vertical profiles of these species have been reported using gradient observations from various measurement techniques such as tethered balloons, unmanned aerial vehicles, towers, remote sensing techniques and aircraft (Benish et al., 2020; Chang et al., 2025; Glaser et al., 2003; Song et al., 2024; Vo et al., 2018; Wang et al., 2025). Among these, Multi-Axis Differential Optical Absorption Spectroscopy (MAX-DOAS) is a cost-effective, automated remote sensing technique capable of observing vertical changes in atmospheric trace gas concentrations over long periods of time. This method has been extensively applied worldwide for atmospheric composition monitoring (Chang et al., 2025; Franco et al., 2015; Jiang et al., 2025; Wagner et al., 2011; Wang et al., 2025).

Previous studies reported that well-established retrieval frameworks for $NO_2$ and HCHO have shown good agreement with satellite and in situ measurements, demonstrating its robustness for long-term and multi-environment observations (Chan et al., 2020; Jiang et al., 2025; Kumar et al., 2020; Martin et al., 2004; Schreier et al., 2020; Schreier et al., 2016; Wagner et al., 2011; Zhang et al., 2022). CHOCHO is challenging to detect due to its weak absorption and spectral interferences, yet MAX-DOAS remains one of the few ground-based techniques capable of reliable retrievals. Successful measurements in both urban and biogenic regions have advanced understanding of CHOCHO photochemistry and have been used to validate satellite products (Hong et al., 2024; Hoque et al., 2018; Lerot et al., 2021; Liu et al., 2021; Schreier et al., 2020; Zhang et al., 2025). Overall, MAX-DOAS demonstrates robust and reliable performance in monitoring $NO_2$, HCHO, and CHOCHO, providing strong methodological support for the data quality in this study.

$NO_2$ concentrations typically decrease exponentially with altitude due to the fact that it originates mainly from surface-level

anthropogenic emissions, whereas photochemical oxidation of long-lived volatile organic compounds (VOCs) at high altitudes sometimes leads to higher distribution heights for HCHO and CHOCHO (Hong et al., 2022; Kumar et al., 2020; Liu et al., 2023). The ratio of CHOCHO to HCHO ($R_{GF}$) serves as a robust indicator for distinguishing VOCs sources due to their different sources and similar sinks (Chan Miller et al., 2014; Vrekoussis et al., 2010; Xing et al., 2020; Zhang et al., 2021). Observations reveal altitude-dependent characteristics of $R_{GF}$, with lower values at low altitudes reflecting dominant biogenic VOCs (BVOC) contributions, while elevated $R_{GF}$ at higher altitudes highlights anthropogenic VOCs (AVOC) (Hong et al., 2022; Xing et al., 2020). The HCHO to $NO_2$ ratio (FNR) is widely used to assess ozone ($O_3$) formation sensitivity (OFS) by classifying regimes into VOC-limited, transition, and $NO_x$-limited (Jiang et al., 2025; Liu et al., 2021; Wang et al., 2025). However, recent studies propose the CHOCHO as a more reliable indicator because it is more directly linked to secondary formation through the VOCs oxidation than HCHO (Hong et al., 2024; Liu et al., 2021). Therefore, comprehensive systematic investigation of vertical distribution patterns of these species is critical for refining OFS assessments across atmospheric layers, optimizing chemical transport models, and informing targeted air quality management strategies.

Coastal atmospheric environments are significantly influenced by sea-land breeze (SLB) circulation and also typhoons in some regions (e.g., the western North Pacific). SLB, a mesoscale phenomenon driven by thermal contrast, exhibits a diurnal alternation of onshore sea breezes (SB) and offshore land breezes (LB), recirculating pollutants between marine and terrestrial zones (Gangoiti et al., 2002; Martins et al., 2012; Zhao et al., 2022). SB enhances pollutant accumulation through reduced boundary layer height (BLH) and suppressed vertical dispersion (Puygrenier et al., 2005), while its moisture influx alters temperature fields and photochemical conditions, complicating secondary pollutant (e.g., $O_3$, secondary aerosols) formation (Gangoiti et al., 2002; Xu et al., 2021). Typhoons are large-scale weather systems that can cleanse pollutants via strong winds and precipitation (Roux et al., 2020; Yang et al., 2012), yet their peripheral subsidence zones with reduced cloud cover intensify solar radiation, triggering severe surface $O_3$ pollution (Deng et al., 2019; Wu et al., 2013). Additionally, typhoon-enhanced winds promote cross-regional $O_3$ transport (Qu et al., 2021; Wang et al., 2024). Both air currents critically modulate pollutant vertical structures and photochemical environments across atmospheric layers. Nevertheless, research gaps persist regarding vertical distributions of $NO_2$, HCHO, and CHOCHO under SB and typhoon conditions, particularly in island where complex topography and unique atmospheric environments may cause these pollutants to behave differently than over continental or

75 coastal regions.

Hainan Island is away far from mainland China and lies approximately 20 km from the nearest point in Guangdong Province. The island spans approximately 32,900 km$^2$ and supports a population of 10.48 million people concentrated in coastal cities (e.g., Haikou, Sanya, and Danzhou), while also maintaining dense vegetation with over 60% forest coverage featuring unique tropical ecosystems (Hainan Provincial Bureau of Statistics, 2024). Adjacent to the South China Sea, its favorable topography

and geographic position render the island frequently influenced by SB and typhoons (Fu et al., 2023; Liang and Wang, 2017; Zhang et al., 2014). Given the good air quality—mean NO$_2$ (7.34 μg m$^{-3}$) and O$_3$ (52.80 μg m$^{-3}$) concentrations below the National Ambient Air Quality Standard (NAAQS) Grade II and urban VOCs (13.1 ppbv) remaining significantly lower than megacities like Beijing and Shanghai—it has earned designation as a National Ecological Civilization Pilot Zone of China (Xu et al., 2024). Notably, several studies have shown that typhoons and the East Asian monsoon can transport pollutants such as

O$_3$ and its precursors to the island via regional-scale advection, thereby inducing pollution events (Fu et al., 2023; Xu et al., 2024; Zhan et al., 2023). However, the specific effects of SB and typhoons on O$_3$ precursors over Hainan Island are not investigated yet.

By combining ground-based MAX-DOAS measurements with meteorological reanalysis data, we investigated the spatiotemporal distributions of NO$_2$, HCHO, CHOCHO, and their photochemical indicators under various air current patterns

(ACPs) in the coastal area of Hainan. This study aims to elucidate the transport mechanisms of pollutants in complex atmospheric environments and their response characteristics to local photochemical processes, thereby supporting air pollution control and environmental management in coastal regions.

## 2 Data and Methods

### 2.1 Measurement of trace gases profile

This study was conducted in Hainan National Field Science Observation and Research Observatory for Space Weather (19.53°N, 109.14°E) at Fuke village (FK), Danzhou City, Hainan Province, during the summer of 2024 (June 1 to August 31). The observation site (altitude approx. 100 m above sea level) lies within a topographically complex region bordered to the

northwest by the Gulf of Tonkin and to the southeast by the Wuzhi Mountain (Fig. 1a). During the observation period, the observation site was directly affected due to the passage of Typhoon "Prapiroon" from south-east to north-west across Hainan Island. The national expressway G98 passes about 1,600 m north-west of FK (Fig. 1b). The MAX-DOAS instrument comprises a scanning telescope, a stepper motor controlling the zenith observation angle, an Ocean Optics QE65 Pro spectrometer, and a data processing computer (Fig. 1c). Data acquisition, with a temporal resolution of 30 seconds for individual spectrum, is conducted daily from 06:00 to 19:00 local time (LT), with dark current signals automatically extracted from nightly background measurements. A full scan sequence of different elevation angles takes approximately 5 minutes to complete.

Spectral data were analyzed using the QDOAS software (http://uv-vis.aeronomie.be/software/QDOAS/), which retrieves differential slant column densities (DSCD) of trace gases by fitting their characteristic absorption bands with reference cross-sections and background corrections. Both ultraviolet (UV) and visible (VIS) channels were employed for oxygen dimer ($O_4$), $NO_2$, HCHO and CHOCHO DSCD retrievals, with configurations detailed in Table S1. Zenith spectra during each scan cycle served as Fraunhofer reference spectra (FRS) to roughly eliminate stratospheric contributions. Measured spectra underwent minor shift and stretch corrections to mitigate instrument instability effects. Quality control excluded retrievals with Root Mean Square (RMS) > 0.001 and DSCD error to DSCD ratios exceeding 10% (30% for HCHO, 50% for CHOCHO). Observations before 07:00 and after 18:00 were excluded due to insufficient light intensity. Detection limits (DDL) for different elevation angles were calculated following S1 (Supplement). $NO_2$ data all exceeded DDL, while HCHO and CHOCHO exhibited significant proportions of sub-DDL values (particularly CHOCHO at high elevations), attributable to their low concentration levels. To preserve temporal continuity, sub-DDL results were retained rather than discarded in subsequent analyses. Typical DOAS fitting results for four gases are illustrated in Fig. S1.

This study employed the HEIPRO algorithm (Heidelberg Profile, developed by IUP Heidelberg) based on optimal estimation methodology (OEM) to retrieve vertical profiles of tropospheric trace gases (Frieß et al., 2006; 2011). The algorithm incorporates the SCIATRAN radiative transfer model (RTM) as an orthorectification mode (Rozanov et al., 2002). The retrieval strategy comprises two sequential steps: (1) aerosol extinction vertical profiles are derived from measured $O_4$ DSCD, followed by (2) trace gas profile retrievals using the aerosol profiles as inputs in RTM. Vertical grids for trace gases span 0.0–3.0 km altitude with 100 m resolution, and retrievals were performed at 10-minute intervals. Exponential decay functions

served as a priori profiles for all species with error $S_a$ of 100% and 500 m covariance correlation length. Fixed aerosol

parameters included: 0.3 km$^{-1}$ extinction coefficient, 0.1 surface albedo, 0.95 single scattering albedo, and 0.72 asymmetry

parameter at 360 nm (Henyey and Greenstein, 1941; Zhang et al., 2021). Surface concentrations of $NO_2$, HCHO, and

CHOCHO were initialized at 2, 5, and 0.2 ppbv, respectively. Representative retrieved profiles are illustrated in Fig. S2. To

ensure that further analyses were based on reliable data, the absolute deviation thresholds between measured and modelled

DSCD (optical densities, OD) values were set to $1 \times 10^{-3}$, $2 \times 10^{15}$, $7 \times 10^{15}$ and $4 \times 10^{14}$ molec cm$^{-2}$ for $O_4$ OD, $NO_2$, HCHO

and CHOCHO DSCD, respectively, in addition to retaining only those results with a degree of freedom of the signal (DoF) >

1.0. Filtered data showed strong model-measurement agreement ($R^2 > 0.95$), confirming retrieval accuracy (Fig. S3).

In addition, hourly surface $O_3$ mass concentrations (μg m$^{-3}$) at the reference state (298 K, 1,013 hPa) were recorded at the

DongPo School monitoring station (DPS, 19.52°N, 109.56°E) using a Thermo Scientific 49i $O_3$ analyzer were converted to

surface VMRs (ppbv) for the analysis of photochemical indictors.

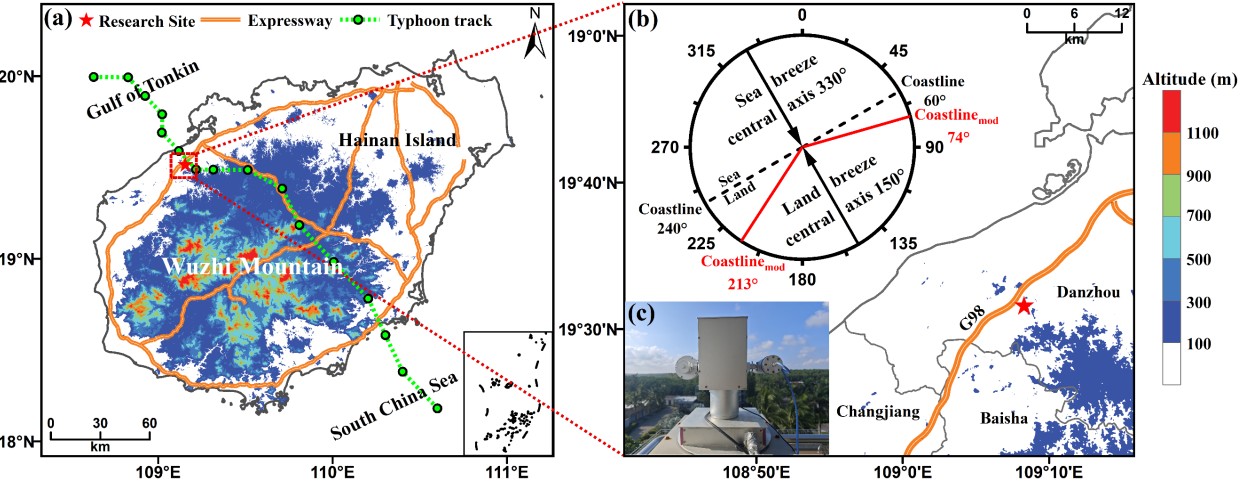

**Figure 1. Overview of the study area and MAX-DOAS. (a) Topographic map of the area around the geopotential height and Multi-Axis Differential Optical Absorption Spectroscopy (MAX-DOAS) measurement station (19.53°N, 109.14°E) in Hainan Province, marked by a red five-pointed star, and the highway network is indicated by an orange line. The track of the typhoon "Prapiroon" is shown by the green dotted line; (b) Topographic map of the area around the observation site, where a highway (G98) passes about 1,600 m to the northwest of the observation site. Compass charts for recording the direction of the sea breezes (SB) and land breezes**

**(LB) are given in the upper left. The central axes of the SB and LB directions are shown by black arrows, and the red solid lines represent the modified coastline (Coastline$_{mod}$); and (c) MAX-DOAS instrument.**

## 2.2 Identification of sea and non-sea breeze days

In this study, a filtering method was adopted as the primary means of identifying days favorable to SB development. Although this approach may underestimate SB frequency, the results will not be ambiguous. Accurate identification of onshore and offshore wind direction (WD) is essential for the method's effectiveness, with the geometry of the coastline serving as the basis for determining the central axes of SB and LB (Azorin-Molina et al., 2009; Azorin-Molina et al., 2011; KiranKumar et al., 2019). In the vicinity of FK, the coastline at larger scales is approximately linear, extending from 240° to 60°, with the central axes of the SB and LB oriented at 321° and 141°, respectively (Fig. 1a and b). However, previous studies have shown that local coastline geometry and topographic influences, in combination with coastal-parallel flows driven by the Coriolis force, can induce deviations in the SB direction (Azorin-Molina et al., 2011; Miller et al., 2003). Based on these local features, the directional range of the SB at FK was revised to span from 213° to 74° (The red solid lines on the compass map in Figure 1b). In addition, islands and peninsulas can trigger SB fronts along different segments of the coastline (Zhu et al., 2017), whereas mountainous terrain often inhibits inland propagation (Barthlott and Kirshbaum, 2013). Specifically, on Hainan Island, the central Wuzhi Mountain range acts as a barrier to the southeastward SB front, limiting its reach into the FK. Notably, during summer, the island is predominantly influenced by the southwesterly monsoon, which frequently produces daytime background winds oriented landward or along the coast (Fig. S4). The formation pattern of these background air currents is fundamentally different from that of the SB. To enhance the contrast with sea breeze days (SBDs), dates influenced by background winds or failing to meet the SB criteria were identified. Those with anomalously elevated daytime relative humidity were excluded to minimize the influence of marine air masses. The remaining dates were classified as non-sea breeze days (NSBDs).

Figure 2 illustrates the flow of an objective recognition algorithm (ORA) for SLB containing four modules and 12 filters. Unless otherwise noted, the data used by ORA are near-surface data. Module 1 filters the large-scale circulation background unfavorable to the formation of SLB, one of the important factors influencing the formation of SLB, through the wind speed (WS) and WD at high altitude (filters 1 to 2) (Arrillaga et al., 2020). Module 2 is the core module, which contains three channels: channels 1 (filter 3 to 7-1) and 2 (filter 3 to 7-2) are based on the sudden change of WS and WD to identify the SB, and the rate of change of WS is introduced to capture the dynamics of the local wind field (filter 5-2), which can make up for

the deficiencies of the traditional WS thresholding method; channel 3 rejects dates potentially affected by oceanic currents by setting a relative humidity (RH) increase threshold (filters 8-9). Module 3 excludes large-scale airflow disturbances through barometric pressure conditions (filters 10-11). Module 4 ensures that the remaining days satisfy the minimum sea-land temperature gradient (filter 12) as a prerequisite for SB establishment. The specific functions of the modules and filters are detailed in S2 (Supplement) and identification results are summarized in Table S3 and Section 3.2.

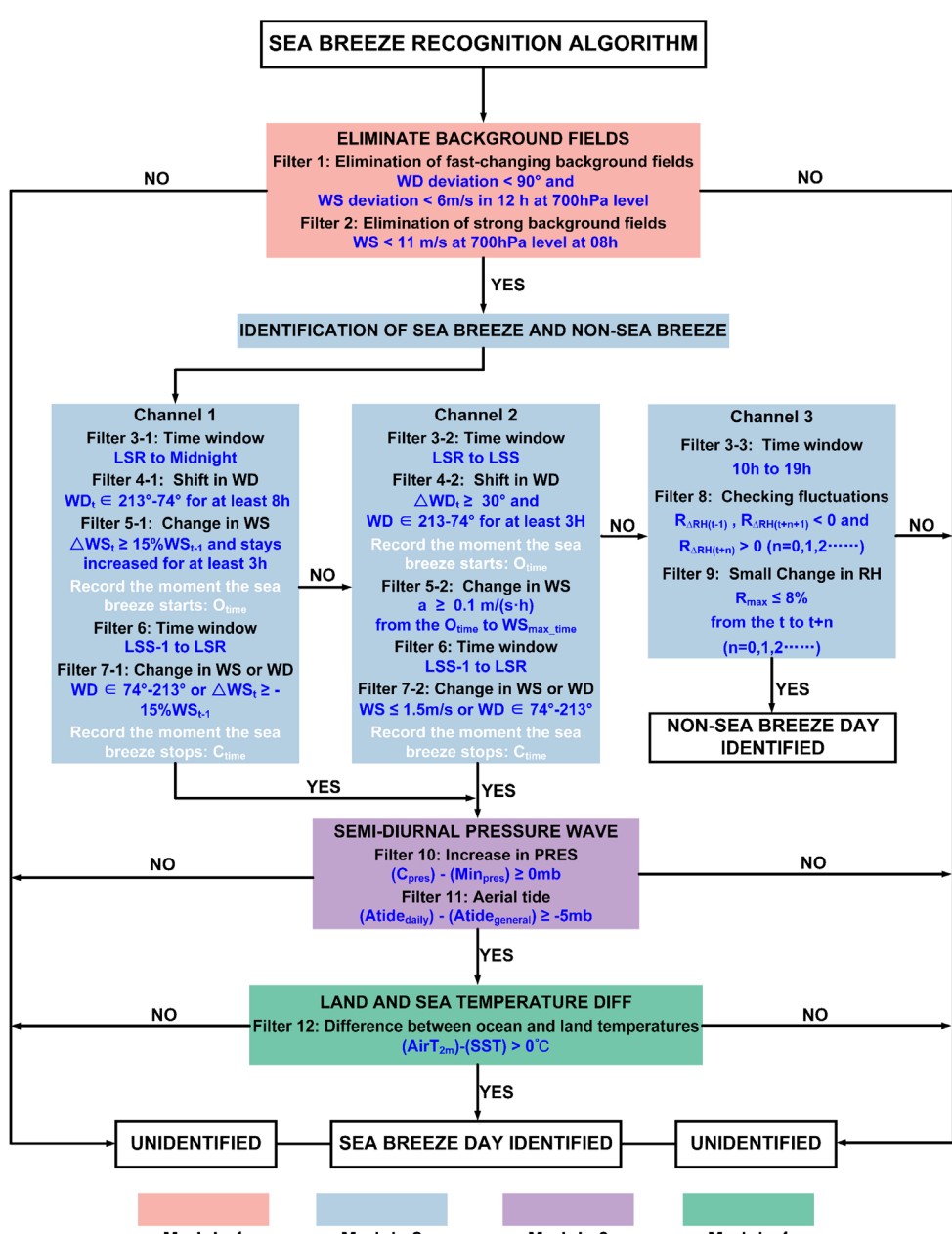

**Figure 2. Flowchart of the objective recognition algorithm (ORA) for SBDs. The pink squares represent eliminate background fields, referred to as Module 1; the blue squares represent identification of sea breeze and non-sea breeze, referred to as Module 2; the purple squares represent semi-diurnal pressure wave, referred to as Module 3; the green squares represent land and sea temperature diff, or Module 4 for short.**

## 2.3 Meteorological reanalysis and ancillary data

Meteorological data were obtained from the European Centre for Medium-Range Weather Forecasts (ECMWF) fifth-generation atmospheric reanalysis product (ERA5), incorporating three principal components: the ERA5 single-level dataset, ERA5 pressure-level data, and ERA5-Land. The single-level dataset provides hourly surface and sea-level parameters at 0.25° spatial resolution, while the pressure-level product delivers three-dimensional atmospheric profiles across 37 vertical levels from 1,000 hPa to 1 hPa, maintaining equivalent spatial and temporal resolutions. The ERA5-Land (0.1×0.1°) component—released in 2019—offers enhanced terrestrial monitoring through refined 9 km spatial resolution while preserving hourly temporal sampling. Specific meteorological variables employed in this analysis, along with their respective dataset sources are comprehensively detailed in Table S4. Notably, ERA5 lacks the surface RH data required; therefore, we computed actual and saturated water vapor pressures using the 2m dew temperature ($T_d$ in K) and 2m temperature (T in K) from ERA5-Land, respectively, and derived RH from their ratio (Zhu et al., 2025):

$$RH = \frac{e_a(T_d)}{e_s(T)} \times 100\% \tag{1}$$

where $e_a(T_d)$ is the actual vapor pressure at dew point temperature, and $e_s(T)$ is the saturation vapor pressure at the current temperature. These can be determined by the following formula:

$$e_a(T_d) = 0.6108 \times exp\left(\frac{17.27 \times (T_d - 273.15)}{T_d - 35.86}\right) \tag{2}$$

$$e_s(T) = 0.6108 \times exp\left(\frac{17.27(T - 273.15)}{T - 35.86}\right) \tag{3}$$

Monthly emission fields of isoprene (0.25°×0.25°), anthropogenic $NO_x$, and VOCs (0.1°×0.1°) were also obtained from the CAMS (Copernicus Atmosphere Monitoring Service) global emission inventories (https://ads.atmosphere.copernicus.eu/datasets/cams-global-emission-inventories?tab=overview).

## 3 Results

### 3.1 Time Series of the Measurement Results

Figure 3 presents the time series of daily averaged near-surface aerosol extinction coefficient (AEC), $NO_2$, HCHO, CHOCHO volume mixing ratios (VMRs), corresponding column integrated aerosol optical depth (AOD) and vertical column densities (VCDs) observed by MAX-DOAS at FK, Hainan Island, China during the 2024 summer season (1 June–31 August). The

measurements reveal distinct atmospheric characteristics shaped by regional environmental conditions. The observed summer aerosol loading (mean AEC and AOD measuring $0.11 \pm 0.03$ km$^{-1}$ and $0.29 \pm 0.1$, respectively) exhibited markedly lower values compared to urban agglomerations like Beijing and Shanghai (where AOD is around 0.4) (Fan et al., 2025; Peng et al., 2025), and slightly higher than in coastal Thailand, where monsoonal rainfall efficiently scavenges aerosols (Peengam et al., 2025). The average $NO_2$ surface VMRs ($1.61 \pm 0.53$ ppbv) was about 3 to 5 times lower than values recorded in major Chinese megacity centers, such as the Beijing–Tianjin–Hebei region ($7.62 \pm 1.39$ ppbv) and the Yangtze River Delta ($7.45 \pm 0.87$ ppbv) (Lou et al., 2025; Ministry of Ecology and Environment of the People's Republic of China, 2024a, b, c), reflecting minimal local traffic and industrial emissions. Notably, HCHO and CHOCHO concentrations ($4.33 \pm 1.07$ ppbv and $0.10 \pm 0.02$ ppbv, respectively) are comparable to values reported for summertime metropolitan regions, such as Beijing (HCHO: 4.41 ppbv) and Guangzhou (CHOCHO: 0.13 ppbv) (Hong et al., 2024). This likely stems from biogenic emissions from surrounding tropical ecosystems compensating for diminished anthropogenic sources. However, the observed HCHO and CHOCHO substantially higher than values reported in less anthropogenically influenced islands and coastal regions, including the coastal area of Mt. Lao, Qingdao, the Galápagos Islands, and Cape Verde (Lawson et al., 2015; Mahajan et al., 2014; Mahajan et al., 2010; Zhao et al., 2024).

At an hourly temporal resolution (Fig. S6), HCHO and CHOCHO exhibited significant positive correlations with elevated temperatures (> 30°C) and reduced humidity (< 60%), particularly during two enhanced photochemical periods: 12–22 June and 1–23 August, when the elevated concentrations persisted from the surface through elevated atmospheric layers. Surface VMRs and VCDs show relatively similar patterns in their time series for all species because of their predominantly low-level distribution. However, during specific episodes (e.g., August 7–10, 2024), pronounced fluctuations in vertical temperature and humidity profiles resulted in a clear decoupling between surface and column measurements for $NO_2$, HCHO, and CHOCHO, which was more pronounced in aerosols as indicated by the weakest correlation ($R^2=0.34$) between surface AEC and AOD (Fig. S7). These results suggest that the vertical distribution pattern of pollutants may involve more complex physicochemical processes that deserve further investigation.

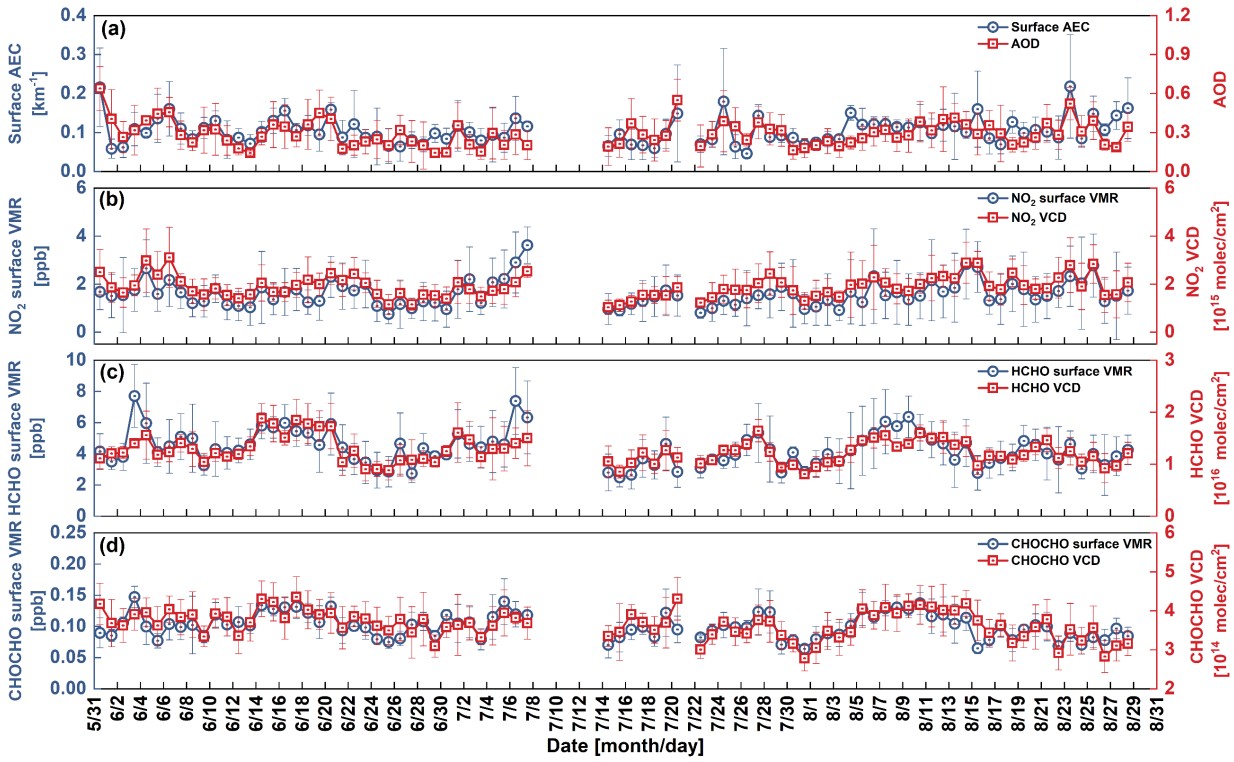

Figure 3. Diurnal variation of pollutants at FK. Time series of daily (a) aerosol, (b) NO₂, (c) HCHO and (d) CHOCHO surface mixing ratios and vertical column densities from MAX-DOAS measurements in Hainan Island, China.

Figure 4 illustrates the average diurnal variation and daily profiles of aerosol, NO₂, HCHO, CHOCHO, temperature and RH during the campaign. Aerosols, HCHO, and CHOCHO exhibited vertical distribution patterns that closely tracked the diurnal variation of BLH. Their vertical distributions peaked synchronously with maximum BLH at noon, followed by gradual subsidence to lower altitudes during afternoon planetary boundary layer (PBL) contraction, though residual amounts persisted in upper layers. The peak of the AEC occurs at the top of the boundary layer almost throughout the observing period due to the aerosols from different sources consistently suppressing the daytime PBL (Xin et al., 2023). Notably, the aerosol vertical distribution bears a remarkable similarity to the RH. The Pearson correlation between AEC and RH remains positive above 0.5 km, reaching a maximum of 0.86 around 1 km, indicating a strong positive relationship at this altitude (Fig. S8). This phenomenon reflects aerosol hygroscopic growth characteristics, particularly the water vapor sensitivity of fine particulate matter, which enhances light extinction at these altitudes (Liu et al., 2020). The occasional observed deviation between surface

AEC and total AOD variations (Fig. 3a) may also be explained by this altitude-dependent hygroscopic behavior.

The diurnal $NO_2$ cycle exhibited a characteristic bimodal pattern, with primary peaks in morning (08:00 LT, VMRs = 2.67 ppbv) and evening (17:00 LT, VMRs = 1.66 ppbv) aligned with traffic rush hours, reflecting dominant vehicular emission sources. Notably, a secondary minor peak emerged at 14:00 LT, potentially driven by regional transport mediated by local circulation patterns (e.g., SB systems) (Fig. 7b). Vertically, $NO_2$ predominantly resides below 300 m. The high HCHO and CHOCHO morning levels are attributed to the oxidation of residual VOCs by OH radicals during daytime and the rerelease from the aerosols (Liu and Wang, 2024; Xing et al., 2020). In contrast, the delayed HCHO peak (10:00 LT, VMRs = 5.14 ppbv) can be attributed to the photochemical oxidation of early morning traffic emitted VOCs under increasing solar radiation. CHOCHO surface VMRs exhibited a distinct peak (0.12 ppbv) between 12:00–13:00 LT, potentially linked to photooxidation of VOCs—particularly alkenes and aromatics—and elevated temperatures that intensified photochemical activity and biogenic isoprene emissions (Duncan et al., 2010; Hendrick et al., 2014). However, a slow decreasing trend regarding the HCHO surface VMRs was still detected in the concurrent period, most likely due to the stronger vertical convection after midday, transferring the secondary HCHO production to higher altitudes (< 800 m) and thus diluting the surface concentration. In contrast, CHOCHO remains confined to lower altitudes (< 500 m) (Fig. 4d, left), mainly because its weaker secondary formation at higher altitudes compared to HCHO and stronger uptake by humid aerosols that depletes its abundance (Fortems-Cheiney et al., 2012; Liao et al., 2025; Sander, 2015; Waxman et al., 2015; Wolfe et al., 2019). The late afternoon rebound in HCHO and CHOCHO concentrations may be due to the combing effects of increased anthropogenic emissions of traffic and cooking activities, as well as reduced atmospheric mixing with temperatures drop and solar radiation decreases (Fig. S9).

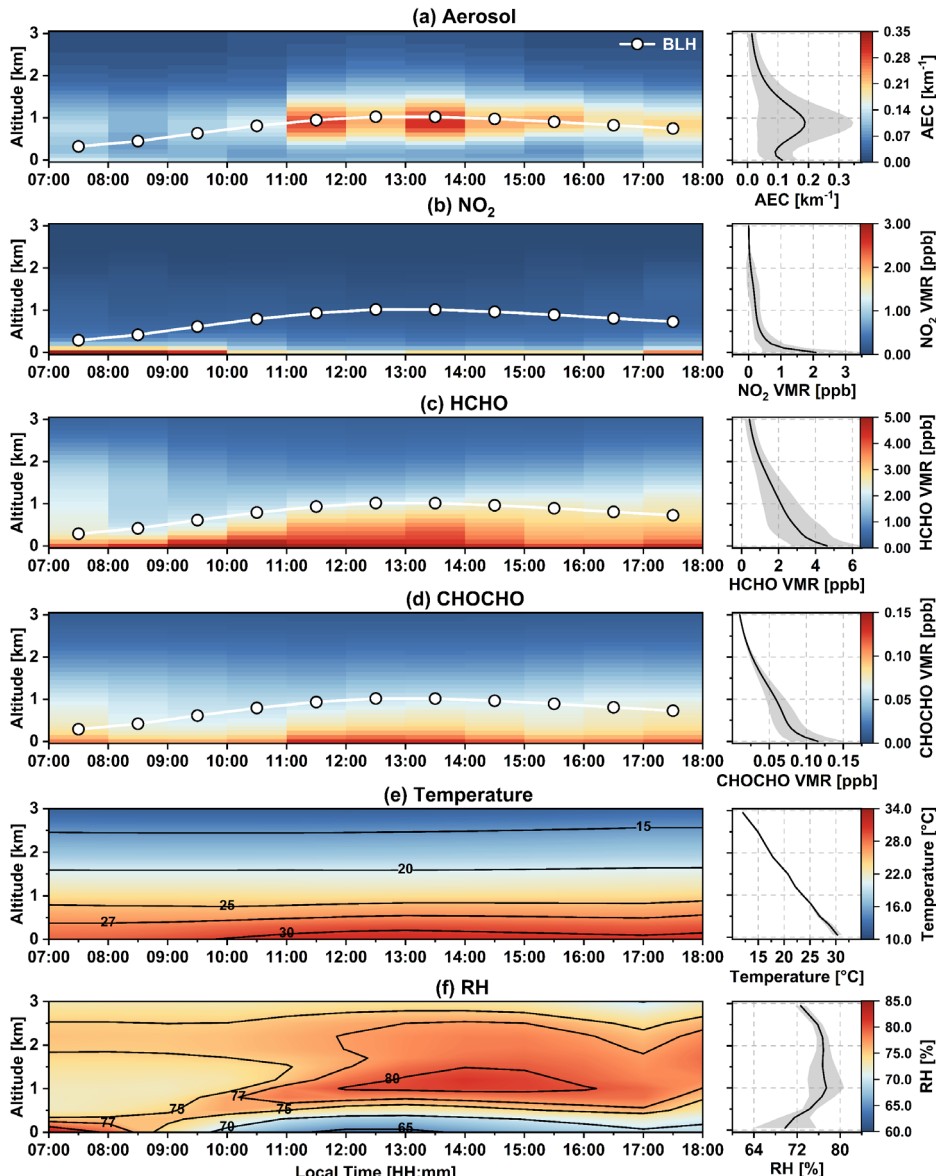

**Figure 4. Diurnal variations in vertical profiles of pollutants and meteorological factors at FK. (a) AEC, (b) NO₂, (c) HCHO, and (d) CHOCHO measured by MAX-DOAS; (e) Temperature and (f) relative humidity (RH) from ERA5 reanalysis. Boundary layer height (BLH) cycles are indicated, while mean vertical profiles displayed in right panels.**

## 3.2 Air current patterns impacted meteorological parameters and trace gases

During the campaign, a unique opportunity arose to study the impact of Typhoon "Prapiroon", which traversed Hainan from southeast to northwest on July 21–23, 2024. Its intense disturbances to the local meteorology continued for one week after

landfall (Fig. 1a and 6). Although the seven typhoon-influenced days (TDs) may be not sufficient sample from a statistical perspective, they represent a coherent case of a recurring meteorological phenomenon—typhoons following a characteristic east-to-west track across the South China Sea. This consistent pathway results in systematic advection of specific air masses into the study region. So we treated these seven TDs as an independent case against NSBDs and SBDs identified by the ORA (Table S3). Days influenced by synoptic-scale systems or hybrid weather patterns were excluded to ensure sample representativeness. Notably, most of the SBDs occur consecutively. This can be explained by the mesoscale circulation, which usually occur under stable atmospheric conditions dominated by persistent high pressure. Figure 5 presents the occurrence frequency of NSBDs, SBDs, and TDs, as well as the frequency distribution of SB onset and cessation times ($O_{time}$ and $C_{time}$). The frequency of NSBDs and SBDs was 32 days (34.8% of the total observation period) and 31 days (33.7%), respectively. The study site exhibited 6.3% lower SB frequency compared to Hainan's regional average (Zhang et al., 2014). SB initiation predominantly occurred after 10:00 LT, peaking at 10:00–13:00 LT (n = 21), likely driven by rapid solar heating in late morning (Parker, 2015). Cessation events clustered post-sunset, with a maximum occurrence at 18:00 LT (n = 16), consistent with the restoration of thermal equilibrium following land surface cooling.

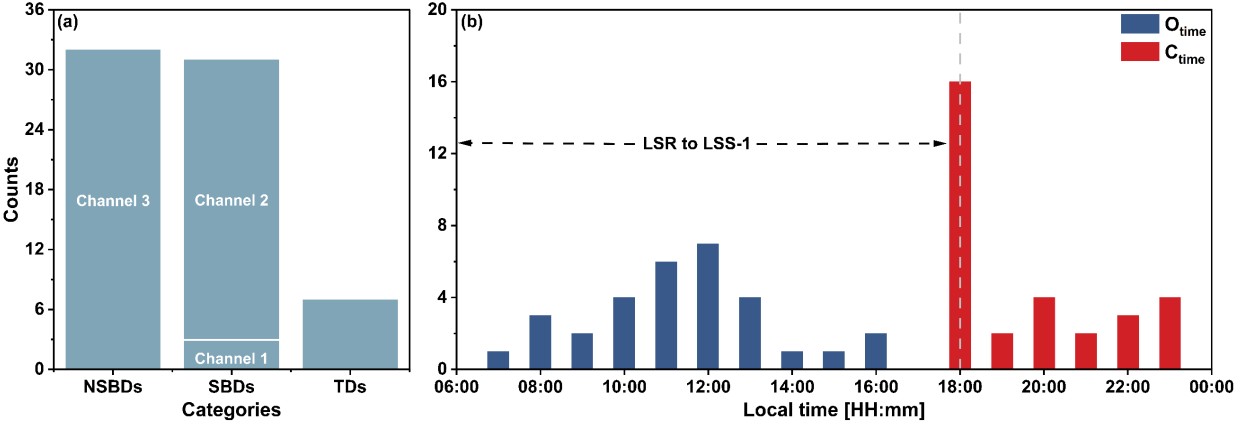

**Figure 5. Statistics of the results of objective recognition algorithm for different air currents. (a) the frequency of occurrence of NSBDs, SBDs and TDs, as well as (b) the frequency distribution of SB onset and cessation times ($O_{time}$ and $C_{time}$). The time ranges for local sunrise (LSR) and one hour before local sunset (LSS) are marked with dotted arrows.**

Afterwards, the mean diurnal variations of each meteorological parameter: WS, WD, temperature, RH, BLH, UV radiation flux (UVF) and vertical velocity ($\omega$) in three ACPs are summarized in Figure 6. It should be noted that the composite analysis

of SBDs excludes dates selected through Channel 1 methodology due to systematic discrepancies in meteorological characteristics compared to Channel 2-derived selections (Fig. S10). This exclusion is maintained in all subsequent analyses unless otherwise specified. Typhoon events show higher WS than NSBDs, but their diurnal variations are similar. The prevailing southwesterly winds are caused by the typhoon's peripheral airflow over the area of measurement, whereas NSBDs

feature daytime southwest winds shifting to southerly at night. The weaker WS of SB (peak 1.89 m s$^{-1}$ at 14:00 LT) likely results from urban/vegetation friction and climate impacts of warming and urbanization (Fu et al., 2011; Guo et al., 2011; Li et al., 2018). Observational data reveal a distinct diurnal cycle in SBDs: after sunrise, WS decreased markedly until reaching a trough at 10:00 LT, followed the WD veered from southeast (149.7°) to south-southeast (170.3°), indicating dissipation of the onshore winds. SB onset at 11:00 LT brought a 41.3° clockwise shift to southwest and rising WS. Maximum SB intensity

occurred at 15:00 LT with northwesterly flow (289.3°), then winds gradually weakened and shifted offshore. A sharp transition (17:00–18:00 LT) featured a 97.2° anticyclonic shift from southwest (222.3°) to southeast (125.0°), with minimal wind speed (0.37 m s$^{-1}$), signaling SB termination and LB onset.

Compared to SB and typhoon conditions, NSBDs exhibit higher surface temperatures and UVF coefficients, likely attributable to reduced cloud cover and enhanced solar irradiance. The well-documented SB thermal mitigation effect manifests itself in

the transport of water vapor inland by advancing oceanic breeze fronts (Yan and Anthes, 1988), producing discernible surface cooling and driving a premature rise and anomalous increase in RH following the SB onset (13:00–17:00 LT) (Fig. 6c,d). Typhoon conditions exhibit thermal minimums due to combined radiative shielding from cyclonic circulation and extensive cloud coverage, while maintaining elevated humidity levels (mean RH: 82.28 ± 3.49%) through continuous moisture advection. BLH displays distinct variability across ACPs: NSBDs show rapid BLH growth peaking at 1,215.72 m (13:00 LT) from

enhanced thermal convection, whereas TDs achieve the highest mean BLH (694.07 ± 122.07 m) through persistent vertical wind shear and turbulent mixing (NSBDs: 619.26 ± 363.76 m, SBDs: 385.31 ± 281.80 m). Consistent with previous findings, SB conditions exhibit suppressed PBL development, attributable to periodic advection of stable marine air masses that inhibit atmospheric destabilization (Li et al., 2025; Puygrenier et al., 2005). As shown in Fig. 6(g) to (i), the differences in BLH further modulate vertical motion under the three ACPs. The positive values of vertical velocity ($\omega$) indicate downward airflow

movement. Upward motion prevails below 2 km on NSBDs and TDs, associated with enhanced convection and typhoon-

induced ascent, respectively. In contrast, SBDs feature weak morning uplift that transitions to subsidence as the SB intensifies, with the strongest downward motion ($\omega \approx 0.05$ m s$^{-1}$) near 1.5 km, suggesting that the SB further restricts pollutant dispersion in FK.

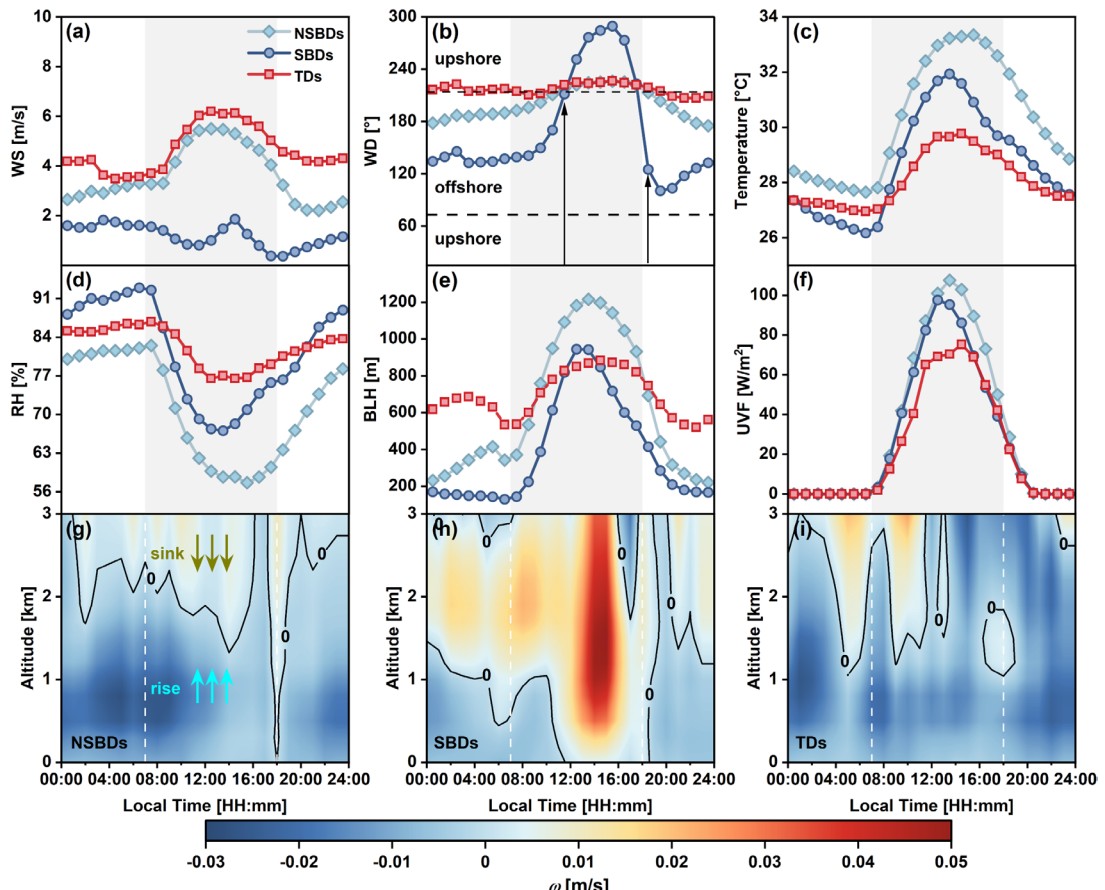

Figure 6. Diurnal variations of meteorological factors under different air currents. Mean diurnal variations of (a) WS, (b) WD, (c) temperature, (d) RH, (e) BLH, (f) UV radiation flux (UVF) and (g-i) vertical velocity ($\omega$) under three air current patterns (ACPs): NSBDs, SBDs, and TDs. The gray shaded area represents the observation period of MAX-DOAS. The two black dotted lines define the range of WD for onshore and offshore winds. Critical transition points for SB initiation and cessation are marked with arrows in (b). The blue and brown arrows represent rising and sinking air currents, respectively.

Accompanied with meteorological parameters, it can be expected that the vertical distribution of NO$_2$, HCHO and CHOCHO would be also varied under different ACPs. As shown in Figure 7, surface NO$_2$ exhibit typical bimodal diurnal variations (morning/evening peaks) during both NSBDs and TDs, in contrast to the distinct accumulation pattern during SBDs, where

concentrations rise sharply from 12:00 LT and peak at 15:00 LT (2.10 ppbv) coinciding with SB front progression and representing a 69.6% overall increase. This synchronization suggests that pollutants from the onshore WD (particularly the northwest and northeast) are transported locally by the SB and accumulate due to the weak WS (Fig. S11). Both HCHO and CHOCHO average surface concentrations exhibited similar gradient patterns along with ACPs: NSBDs > SBDs > TDs, which is closely related to the differences in meteorological factors (WS, temperature, RH and UVF) (Fig. 6). Diurnal profiles of surface HCHO and CHOCHO generally followed their average patterns throughout the summer, though SBDs induce a unique bimodal HCHO distribution and a delayed CHOCHO peak compared with NSBDs and TDs, which may be attributed to sea-breeze circulation transports VOCs-rich air masses from inland and enhanced secondary formation of HCHO and CHOCHO. Vertical profiling analysis reveals distinct stratification characteristics of pollutants across three ACPs. For $NO_2$, maximum concentrations remain consistently confined below 300 m altitude across all patterns. Similar to the surface concentrations, the vertical gradients of HCHO and CHOCHO followed NSBDs > SBDs > TDs sequence, clustering at 800/500/300 m and 400 m/300 m/surface, respectively. Thermodynamic vertical transport mechanisms significantly influence pollutant distribution. Intense midday thermal plume convection causes HCHO and CHOCHO to disperse upward. As temperatures drop, convective activity decreases, but residual pollutants remain at higher altitudes and continue vertical transport through atmospheric motions. During marine breeze episodes, the thermal internal boundary layer (TIBL, ~200 m thick) forms as cold marine air flows over the warm land surface, restricting pollutant dispersion (Arrillaga et al., 2016; Cuxart et al., 2014; Talbot et al., 2007). Observations revealed that pollutant concentrations were significantly elevated within the TIBL during SB penetration, with CHOCHO being particularly persistent. Interestingly, pollutant concentrations aloft decreased markedly around 15:00 LT, particularly for HCHO, likely due to the combined effects of strong subsidence and a concurrent shift in upper-level WD that weakened pollutant transport (Fig. 6h and S12).

Surface VMRs to VCDs ratios ($R_{SV}$), quantifying altitudinal distribution patterns (Fig. 7d,h,i). Elevated $R_{SV}$ values indicate enhanced near-surface accumulation, whereas lower values suggest predominant vertical dispersion to higher altitudes. Following SB incursion, the universal elevation of $R_{SV}$ across species suggests that the onshore winds primarily affect pollutants in the lower atmosphere. Notably, the significantly more pronounced $R_{SV}$ enhancement observed in CHOCHO relative to HCHO suggests that marine-transported VOCs prefer to produce CHOCHO rather than HCHO. During typhoon

events, persistent updrafts transport pollutants to higher altitudes, resulting in the lowest mean $R_{SV}$ values, particularly for HCHO, which remains present between 1 and 3 km. Overall, favorable meteorological conditions on NSBDs promote photochemistry and vertical mixing, while SB transport and strong subsidence enhances near-surface pollution levels. Typhoons dilute pollutants but lift them to higher altitudes.

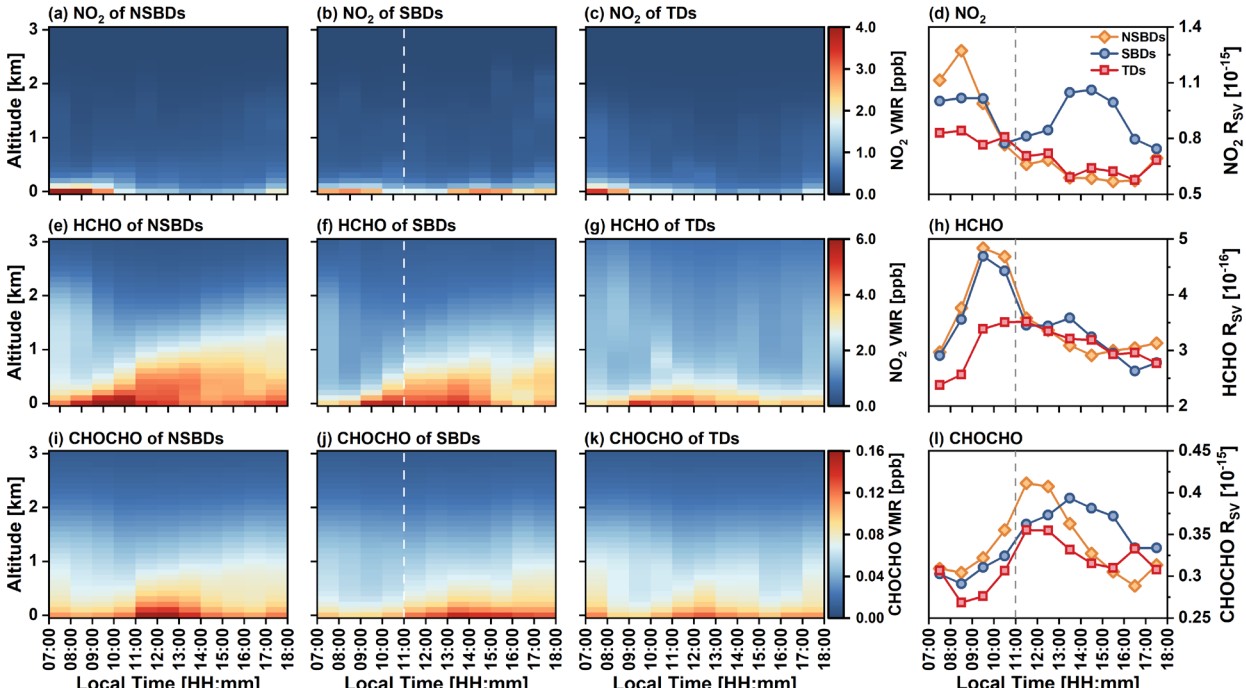

**Figure 7. Diurnal variations in vertical distributions of pollutants under different air currents. Mean diurnal vertical distribution of (a-c) NO₂, (e-g) HCHO, and (i-k) CHOCHO on NSBDs, SBDs, and TDs. Diurnal variations of the ratios of surface VMRs and VCDs ($R_{VS}$) for (d) NO₂, (h) HCHO and (l) CHOCHO are shown in the far right column. The $O_{time}$ of the SB is indicated in the SBDs and $R_{SV}$ panel by a white dashed line and a grey dotted line, respectively.**

### 3.3 Ozone and photochemical indicators under different air currents

To investigate the influence of different ACPs on summertime $O_3$ concentration and photochemical formation, K-means clustering was to characterize diurnal pattern of $O_3$ concentration (S3 (Supplement)). Figure 8 shows the daily profiles of $O_3$ and its precursors (NO₂ and HCHO from MAX-DOAS) and meteorological parameters (temperature, RH and total precipitation (TP)) for the five categories during the observation period. The difference between the midday peak and the morning minimum represents net $O_3$ increment ($\Delta O_3$). The statistics on the number of days, background levels, $\Delta O_3$, NSBDs

and SBDs for each category are shown in Table 1. Five $O_3$ clusters exhibit different concentration characteristics: Cluster 1 (moderate background/low $\Delta O_3$), Cluster 2 (low background/high $\Delta O_3$), Cluster 3 (low background/extreme $\Delta O_3$), Cluster 4 (high background/low $\Delta O_3$), and Cluster 5 (low background/moderate-high $\Delta O_3$). Moreover, Cluster 1 and Cluster 5 exhibit a high prevalence of SBDs, accounting for 64.7% and 83.3%, respectively, while Clusters 3 and 4 are predominantly composed of NSBDs, at 71.4% and 86.7%. Cluster 2 shows an equal distribution of 50% each.

**Table 1. After dividing the daily $O_3$ changes into five clusters based on the K-means method, the number of days, the $O_3$ background level, the net $O_3$ increment ($\Delta O_3$), and the number of NSBDs and SBDs for each cluster are as follows.**

| Cluster | days | background level | $\Delta O_3$ | NSBDs | SBDs |
|---|---|---|---|---|---|
| Cluster1 | 17 | 15.31 ppbv | 9.50 ppbv | 6 | 11 |
| Cluster2 | 12 | 9.04 ppbv | 23.43 ppbv | 6 | 6 |
| Cluster3 | 7 | 9.31 ppbv | 35.29 ppbv | 5 | 2 |
| Cluster4 | 15 | 22.23 ppbv | 10.00 ppbv | 13 | 2 |
| Cluster5 | 12 | 7.36 ppbv | 15.81 ppbv | 2 | 10 |

$O_3$ concentrations are influenced by meteorological factors such as temperature, RH, solar radiation and wind. In general, strong solar radiation, high surface air temperatures and low RH favor the photochemical production of $O_3$, leading to increased $O_3$ concentrations (Seinfeld and Pandis, 2016). Specifically, in Clusters 1 and 5, a higher proportion of SBDs results in lower temperatures and elevated RH, leading to significantly reduced overall $O_3$ levels ($20.1 \pm 3.3$ and $15.3 \pm 5.3$ ppbv). In contrast, Cluster 3 and 4, dominated by NSBDs, experience higher temperatures and lower RH, which favor $O_3$ formation, thereby yielding higher $O_3$ concentrations ($25.4 \pm 10.7$ and $27.3 \pm 3.9$ ppbv). Specific daily variations can also be seen, following the arrival of marine air, the decrease in $O_3$ concentration in Clusters 1 and 5 is more pronounced than that in Cluster 3 and 4, occurring nearly in tandem with decreases in temperature and increases in RH, suggesting that the meteorological shifts induced by the SB are generally unfavorable for $O_3$ formation (Fig. 8a,b,c). The correlations between $\Delta O_3$ and both change in temperature ($\Delta T$, defined as the difference between the daytime maximum and the morning minimum) and change in RH ($\Delta RH$, defined as the difference between the morning maximum and the daytime minimum) further corroborate this conclusion (Fig. S14). Compared to NSBDs, $\Delta T$ exhibited a weaker positive correlation ($R^2 = 0.24$) with $\Delta O_3$ during marine breeze periods while maintaining a consistent negative correlation ($R^2 = 0.38$) with $\Delta RH$, suggesting that SB arrival suppressed photochemical $O_3$ production.

Clusters 1 and 4 exhibit relatively low daytime $\Delta O_3$ because both start from elevated nocturnal $O_3$ baselines. In Cluster 1, possible compensation by onshore winds sustained moderate $O_3$ levels despite heavy nighttime precipitation. Lower nocturnal RH in Cluster 4 inhibited plant stomatal uptake and diminishes dry deposition efficiency, allowing $O_3$ to be accumulated (Kavassalis and Murphy, 2017). In contrast, Cluster 5 shows higher $\Delta O_3$ than Clusters 1 and 4, despite its worse meteorological conditions, for two synergistic reasons: first, intense precipitation and high RH suppress the nocturnal $O_3$ baseline, and second, daytime SB transport delivers greater $NO_2$ precursor loads, enhancing photochemical $O_3$ production and amplifying the daytime peak. Cluster 3 features a sharp afternoon increase in $O_3$—reaching 44.6 ppbv at 18:00 LT—likely driven by elevated midday HCHO concentrations boosting radical production, combined with a weakening $NO_2$ titration sink as $NO_2$ levels fall toward evening (Fig. 8a,e,f). The contribution of ACPs to pollutant was quantified using the cumulative ratio ($R_{SN}$) of pollutant concentrations between SBDs and NSBDs across five clusters (Fig. S15). $R_{SN}$ and cumulative $R_{SN}$ approaching 1 or 5 indicate equivalent contributions, while values exceeding or falling below these thresholds demonstrate SBDs or NSBDs dominance, respectively. During 11:00–15:00 LT, $O_3$ $R_{SN}$ for Clusters 1 and 5 (characterized by low temperature, high humidity and high TP) remained below 1, whereas Clusters 2 and 4 (high temperature, low humidity, very low TP) frequently exceeded 1. Furthermore, $O_3$ cumulative $R_{SN}$ gradually rose after the $O_{time}$, marginally surpassing 5 between 14:00–16:00 LT— considerably weaker than $NO_2$. These phenomena suggest that, under favorable meteorological conditions (Clusters 2 and 4), SB-transported precursors (especially $NO_2$) bolster $O_3$ formation, whereas under the typically induced adverse conditions (Clusters 1 and 5), SB prefers to inhibit $O_3$ production.

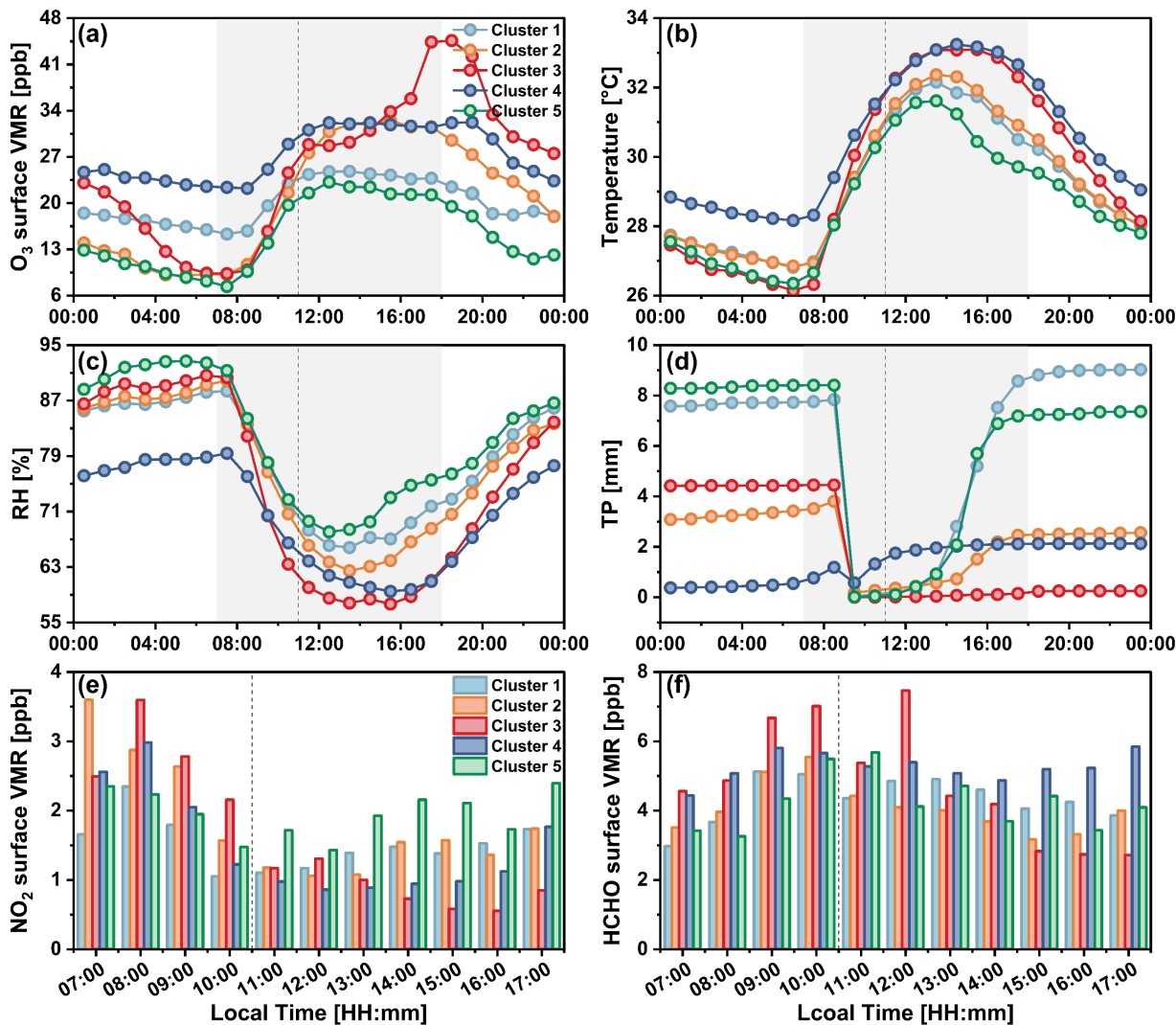

**Figure 8. Diurnal variations of O₃, meteorological factors and O₃ precursors in different categories. Mean diurnal cycles of (a) O₃, (b) temperature, (c) RH, (d) total precipitation (TP), (e) NO₂ and (f) HCHO for five categories. MAX-DOAS instrument observation times are indicated by gray shaded areas. The black dotted line indicates when the SB started.**

Figure 9 illustrates the diurnal variations in the vertical distributions of $R_{GF}$, FNR, and GNR under different ACPs, as well as the average characteristics of the vertical profiles. The average surface $R_{GF}$ for three ACPs were $0.024 \pm 0.0044$, $0.0267 \pm 0.0056$, and $0.0271 \pm 0.0044$, all remaining below the 0.04 threshold even during peak daytime concentrations, matching observational records from rural environments (Lee et al., 1995; Moortgat et al., 2002). On NSBDs, enhanced solar radiation

at midday increased CHOCHO concentrations, leading to a simultaneous increase in $R_{GF}$ at the surface (peak 0.03 at 12:00 LT) and in the altitude layer below 300 m. Owing to the distinct vertical distributions of HCHO and CHOCHO, $R_{GF}$ values observed above 300 m decline markedly, and the low-$R_{GF}$ region ($R_{GF} < 0.025$) progressively migrates upward over time (Fig. 7e,i and 9a). The surface $R_{GF}$ exhibits a markedly different pattern on SBDs. Following the onset of the SB, $R_{GF}$ increases steadily until a surface maximum of 0.038 is reached at 16:00, which again proves that the SB transports more VOCs that tend to produce CHOCHO. Aircraft-based photochemical tracer studies indicated that under low-$NO_x$ conditions, isoprene peroxyl radicals (ISOPO2) rapidly isomerize to form CHOCHO (Chan Miller et al., 2017). Given the relatively low local $NO_2$ levels, it is reasonable to infer that the VOCs introduced by the SB are BVOC (Especially isoprene). The upper $R_{GF}$ also shown a similar upward trend to the near-surface, but is limited to below 300m, suggesting that oceanic air advection transports BVOC mainly in this altitude range (Fig. 9b).

Previous studies have identified regions with biogenic emissions typically showing lower $R_{GF}$ values ($< 0.04$), while higher $R_{GF}$ ($> 0.04$) are associated with strong AVOC emissions (Chan Miller et al., 2014; Chen et al., 2023). As shown in Fig. 9d, all ACPs demonstrated mean $R_{GF}$ values $< 0.04$ across altitudes, indicating BVOC dominance in the FK, which can be supported by the bottom-up emission maps from CAMS reanalysis data (Fig. S16). It shows higher emission flux of isoprene compared to anthropogenic $NO_x$ and VOCs—especially considering that other BVOC species were not included here, e.g., monoterpenes and sesquiterpenes. On NSBDs, $R_{GF}$ progressively increased with altitude, suggesting higher-altitude VOCs preferentially form CHOCHO over HCHO, consistent with previous findings (Hong et al., 2024; Zhang et al., 2021). The overall higher $R_{GF}$ at 0–2,000 m on SBDs before 10:00 LT may be due to a combination of southerly onshore winds and mountain winds in the morning, which bring the BVOC that has accumulated at high altitude during the night from the mountains to the measurement site. The 08:00 LT $R_{GF}$ values are high throughout the vertical column, peaking at 700 m (0.042), especially below 1,000 m where 54.54% of the $R_{GF}$ exceeds 0.04, suggesting that AVOC may be contributing more here. Due to the transport of BVOC emitted from the vegetation of Wuzhi Mountain by "Prapiroon" to FK, the $R_{GF}$ remained elevated at most heights (especially at the mid-levels) under typhoon conditions, with a peak of 0.038 (at the 800 m). Both SB and typhoon scenarios amplified mid-altitude $R_{GF}$, demonstrating distinct vertical profiles from NSBDs.

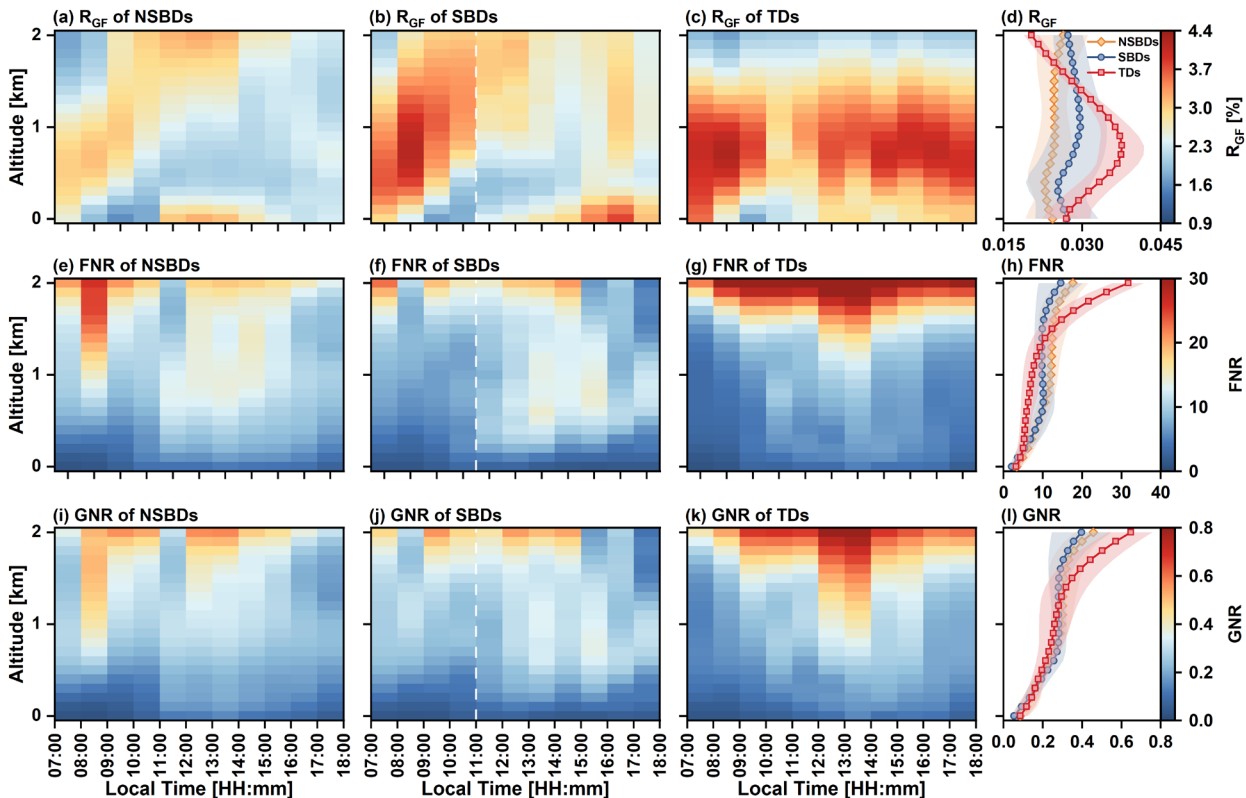

**Figure 9. Diurnal variations in vertical distributions of Photochemical indicators. Mean diurnal cycles in the vertical distribution of (a-c) $R_{GF}$, (e-g) FNR, and (i-k) GNR on NSBDs, SBDs, and TDs. Vertical profiles of (d) $R_{GF}$, (h) FNR and (l) GNR are shown on the far right, with the shaded area indicating the standard deviation. The $O_{time}$ of the SB is indicated by a white dashed line in the SBDs panel.**

Different ACPs have a strong influence on the spatial and temporal distribution of $O_3$ precursors, especially in SBDs and TDs, where the influx of external precursors can lead to anomalous distributions of pollutant concentrations in the near-surface and upper atmosphere, which may alter the regional $O_3$ production mechanism (Fig. 7 and 9). The determination of FNR and GNR thresholds is critical for discerning this regime. In this study, a third-order polynomial fit was employed to elucidate the relationship between FNR, GNR, and $O_3$ concentration (Jin et al., 2020; Ren et al., 2022). Figure 10 shows the strong correlations between $O_3$ levels and both FNR ($R^2 = 0.94$) and GNR ($R^2 = 0.84$) indicate robust polynomial model performance. We treat the transition state as spanning the first 1% of the indicator range of the $O_3$ peak, and end up with local near-surface transition zones of FNR and GNR of 6.33–8.90 and 0.16–0.24, respectively. it should be noted that the fitted curves show only

a slight downward trend when the FNR (7.52) and GNR (0.19) are larger than the indicator values with peak $O_3$ concentrations, and are based on the sparse data, introducing considerable uncertainty. Nevertheless, it is certain that the local thresholds for FNR and GNR are much higher than those reported in previous studies (Liu et al., 2021; Wang et al., 2021). This suggests that the critical importance of localized evaluation for FNR and GNR under conditions of extremely low $NO_2$ concentrations.

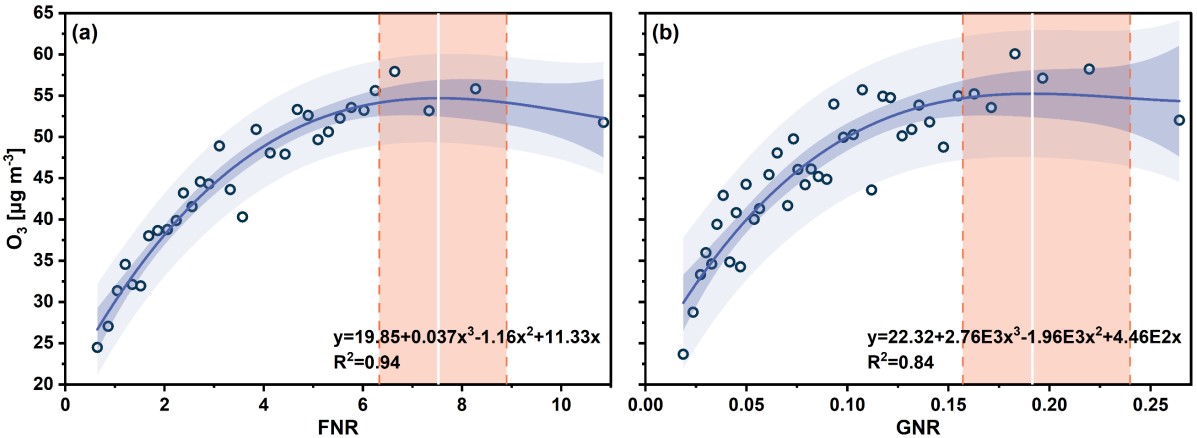

**Figure 10. Diagnosing the $O_3$ formation sensitivity. The relationship between local $O_3$ surface concentrations and (a) FNR and (b) GNR values was fitted based on MAX-DOAS observations. The blue solid line is the fitted third-order polynomial curve, and the shading indicates the 95% confidence level and 95% prediction band, respectively. The white solid line indicates the peak of the fitted curve and the pink vertical shading indicates the range above the top 1% of the fitted curve (transition regime).**

Following the OFS thresholds determined, mean surface FNR and GNR for NSBDs (3.92 ± 1.43 and 0.096 ± 0.044), SBDs (2.53 ± 0.82 and 0.065 ± 0.016), and TDs (3.62 ± 1.33 and 0.097 ± 0.037) indicated that the local near-surface was VOC-limited regimes throughout the day. The FNR and GNR under different ACPs revealed pronounced spatiotemporal variations (Fig. 9e-l). On NSBDs and TDs, two indicators follow a synchronized diurnal cycle, rising in the morning to a maximum at 13:00 LT before declining in the afternoon. In contrast, both ratios on SBDs peak synchronously at 12:00 LT (FNR = 3.89,

GNR = 0.098), and were then rapidly reduced by the continuously transported $NO_2$ during the developmental phase of the SB (12:00–15:00 LT), thus strengthening VOC-limited regimes. With the attenuation of SB, reduced external transport and photochemical depletion led to a decline in $NO_2$ levels, FNR and GNR gradually increase, resulting in a slight shift of the OFS toward the transition regimes, but remained VOC-limited regimes. A mechanism similar to the dual regulation of $O_3$ production by near-surface SBs still applies at lower atmospheric levels (< 300 m), but it is complicated by the weakening of the $NO_2$

effect of SB transport with increasing altitude as well as by the heterogeneity of meteorological factors in the vertical direction. The FNR and GNR above 2.5 km on TDs were much higher than those of NSBDs and SBDs throughout the day.

Since the effect of primary emissions on $NO_x$ concentrations decreases with increasing altitude, the sensitivity of $O_3$ formation to $NO_x$ may increase, leading to an overall increase in FNR and GNR with altitude for all ACPs (Fig. 9h,l). Specifically, less primary emissions and wind dilution on TDs resulted in an exponential increase in both metrics with altitude. In particular,

above 2.5 km, the FNR and GNR are much larger than those of NSBDs and SBDs throughout the day (Fig. 9g,k). The SB inhibited the vertical transport of pollutants, resulting in a reduced sensitivity of the $NO_2$ concentration response at high altitude to anthropogenic emissions, so that above 600 m, the FNR is lower on SBDs than on NSBDs, and the difference progressively widens with increasing altitude. This vertical difference feature is weaker in GNR, which is attributed to the fact that CHOCHO is mainly dominated by the oxidation pathway of VOCs rather than the transport process, compared to HCHO. Below 900 m,

the photo-oxidation of VOCs and its more significant vertical diffusion advantage over $NO_2$ together drive the sharp increase in FNR and GNR with altitude. However, from 900 to 1,500 m, both of them show a stabilizing trend, reflecting the synergistic effects of VOCs and $NO_x$ in this altitude range. At higher altitudes, both metrics again showed an increasing trend due to further attenuation of $NO_x$ concentrations.

To further investigate the distribution of OFS across altitudes under different meteorological patterns, we grouped the two

metrics below 1.2 km altitude based on local BLH levels and assumed that the same OFS thresholds apply at high altitude and near the ground. Figure 11 shows the contributions of the VOC-limited, transition, and $NO_x$-limited regimes in each altitude layer on NSBDs, SBDs and TDs. Overall, surface OFS is predominantly VOC-limited regime (contribution >70%), while at higher elevations its influence decreases and the transitional and $NO_x$-limited regime become dominant. On SBDs, the contribution of FNR-based transition increases from the 600–800 m layer upward, a trend not seen with the GNR indicator,

likely due to differing vertical sensitivities of the two indices to precursors. Below 400 m, the SB inputs more $NO_2$, resulting in a contribution of more than 50% to the VOC- limited regime in the 0–200 m and 200–400 m layers. Notably, GNR identifies transition and $NO_x$-limited regimes more often than FNR. On SBDs, it shows that layers above ~400 m are largely $NO_x$-limited. On TDs, FNR indicates VOC-limitation for most layers except 1000–1200 m, whereas GNR shows transition or $NO_x$-limited regimes above ~200 m. An interesting feature is that both metrics exhibit VOC-limitation aloft on TDs, likely because upper-

level winds align with the SB pathway and transport upstream NO$_x$ into the region. Such behavior shows that WD may be a

key factor driving OFS transitions across different ACPs (Fig. S12). SB and typhoon events transport BVOC and increase

VOCs supply, which theoretically shifts OFS toward NO$_x$-limited or transition regimes—a shift GNR captures more clearly,

while FNR is less sensitive. Prior work shows that elevated primary HCHO can raise FNR independent of photochemistry (Liu

et al., 2021). These findings suggest that (1) GNR may be more suitable as a photochemical indicator of OFS in BVOC-

enhanced or -dominated regions than FNR; (2) reduction of VOCs emissions (especially BVOC) can effectively control the

local surface O$_3$ pollution; (3) NO$_2$ transport barriers need to be set up below the low elevation in response to the SB, to inhibit

the disruption of exogenous pollutants to photochemical equilibrium of this layer; and (4) The formation process of O$_3$

pollution due to regional and vertical transport of precursors under extreme weather is more complicated, and needs to be

combined with regional joint prevention and control and vertical pollution control measures.

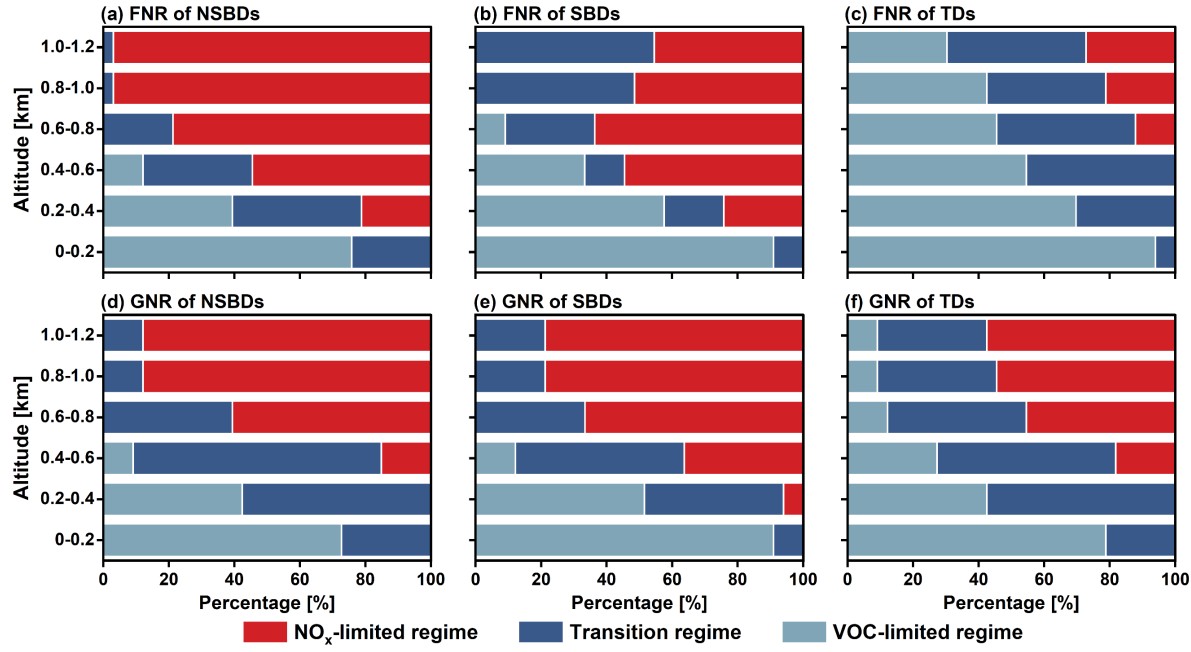

**Figure 11. Vertical distributions of ozone formation sensitivity under different air currents. The contribution of VOC-limited, transition, and NO$_x$-limited regimes for each layer below 1.2 km on (a, d) NSBDs, (b, e) SBDs and (c, f) TDs at the observation site.**

# 4 Discussion

This study employed MAX-DOAS observations to characterize the vertical distribution of daytime aerosols, $NO_2$, HCHO, and CHOCHO in a coastal rural area of an island environment. The aerosol AOD ($0.29 \pm 0.1$) and $NO_2$ surface VMRs ($1.61 \pm 0.53$ ppbv) were substantially lower than those typically observed in urban settings, whereas the HCHO and CHOCHO surface VMRs ($4.33 \pm 1.07$ and $0.10 \pm 0.02$ ppbv) were comparable to those in several megacities (Hong et al., 2024). HCHO and CHOCHO were closely related to high temperatures and low RH, but due to dilution caused by strong vertical convection, surface HCHO did not show signs of significant elevated midday concentrations similar to those of CHOCHO. The hygroscopic effect of the aerosol caused its vertical distribution pattern to be consistent with the RH. $NO_2$, HCHO and CHOCHO were concentrated below 300 m, 800 m and 500 m, respectively. Notably, pollutants remaining at higher elevations in the afternoon were transported further upward.

The observational findings provide critical insights into the characteristics of pollutants on the NSBDs, SBDs and TDs. In this study, an ORA was established for selecting the first two ACPs. The results demonstrate that the occurrence frequencies of SBDs and NSBDs during local summer reach 33.7% and 34.8%, respectively, with SB typically initiating at 11:00 LT and dissipating by 18:00 LT. Indeed, The ORA can be broadly applied, as parameter tuning and threshold adjustment allow for the selection of circulation patterns that correspond to local characteristics. Nevertheless, precise delineation of NSBDs remains challenging in complex island environments under monsoon influence, as transient cloud cover may introduce anomalies in RH variation indices even on valid NSBDs. Combining the ORA with MAX-DOAS observations, we analyzed the spatial and temporal distributions of pollutants and photochemical indicators ($R_{GF}$, FNR and GNR) under different ACPs. The overall low $R_{GF}$ values ($< 0.04$) for all ACPs indicated predominant local BVOC contributions. Elevated concentrations (e.g., $O_3$) and vertical distribution (e.g., HCHO, CHOCHO) of pollutants are observed on NSBDs, driven by highly favorable meteorological conditions that enhance VOCs photochemical oxidation and vertical dispersion, necessitating stringent controls on precursor emissions and transport process under this pattern. SB introduced moist air masses that cooled surface temperatures and its vertical structure reduced BLH, suppressing pollutant generation and dispersion. Notably, SB intrusion enhanced $NO_2$ and BVOC transport below 300 m, which may enhance $O_3$ production under suitable conditions. Typhoons minimized pollutant concentrations through strong dilution by wind and precipitation, yet simultaneously elevated mid-altitude BVOC via regional

transport and promoted vertical dispersion through intense turbulence.

Vertical heterogeneity in OFS under distinct ACPs reveals complex dependence of $O_3$ formation on airflow-driven precursors. Local surface $O_3$ formation showed VOC-sensitive characteristics based on FNR and GNR, transitioning to $NO_x$-limited regime with altitude, but VOC-limited regime still contributed to OFS at high altitude on TDs. Additional BVOC inputs from SB and typhoons and strong local biological discharges suggest that GNR may be a more appropriate indicator of OFS. It should be noted that our determined FNR and GNR thresholds significantly exceeded literature values, suggesting existing classification criteria may inadequately represent tropical coastal rural environments with low $NO_2$ baseline concentrations.

Overall, abatement of VOCs is an effective means to manage local $O_3$, while the basic strategy of controlling the emission of $NO_x$ as well as VOCs from the SB may help to reduce local photochemical pollution. It should be noted that we cannot accurately determine the OFS in the vertical direction because the DPS only provides surface $O_3$ concentration data and carry over the surface OFS thresholds to other altitudes introduces a large uncertainty. Future work should aim to develop local FNR and GNR thresholds in the vertical direction for more accurate assessments.

This study reveals several summertime physical and chemical processes that are characteristic of Hainan Island, so the quantitative values reported here—such as pollutant concentrations, ORA parameter ranges and FNR/GNR thresholds—are shaped by the island's unique land–sea configuration, monsoon regime, and emission environment and should be treated as site-specific. However, the vertical stratification mechanisms linked to species lifetimes and atmospheric dynamics, the thermodynamic and compositional effects of SB intrusions, the typhoon-related scavenging, redistribution, and uplift, and the diagnostic value of different photochemical indicators under different ACPs are processes that commonly occur across many coastal and island settings. Thus, while specific numerical thresholds may vary, the underlying conceptual processes and the ORA framework offer a transferable basis for interpreting similar coastal phenomena, particularly in regions with comparable climatic and geographic conditions.

**Data available.** The MAX-DOAS datasets generated in this study are available from the corresponding authors (S.W.) upon reasonable request for research purposes. $O_3$ data can be downloaded from the https://www.mee.gov.cn, ERA5 pressure-level data can be downloaded from https://cds.climate.copernicus.eu/datasets/reanalysis-era5-pressure-levels?tab=overview, ERA5

single-level data can be downloaded from https://cds.climate.copernicus.eu/datasets/reanalysis-era5-single-levels?tab=overview, ERA5-Land data can be downloaded from https://cds.climate.copernicus.eu/datasets/reanalysis-era5-land?tab=overview.

**Author Contributions.** B. L., S. Z., and Z. J. set up and operated the MAX-DOAS instrument, and B. L. analyzed the data with the help of S. Z., Y. S., and Z. J.; B. L. wrote the manuscript, S. W., B. Z., S. Z., and R. X. provided comments, and S. W. conceived, supervised, reviewed and edited the paper. All authors participated in the discussion of the results and revision of the manuscript.

**Competing interests.** The authors declare that they have no competing interests.

**Acknowledgements.** This study has been supported by National Natural Science Foundation of China (42375089, W2411028), National Key Research and Development Program of China (2023YFC3709401 and 2023YFC3709400), and National Civilian Space Infrastructure Project (Grant No. Y5BZ31AC60). The authors would like to thank the Chinese Ministry of Ecology and Environment for providing the $O_3$ data and to the developers of the ERA5 data for providing free and Open-access materials. The authors would like to thank the Institute of Remote Sensing/Institute of Environmental Physics, University of Bremen, Germany, for providing the SCIATRAN radiative transfer model used in this paper.

**Financial support.** This study has been supported by National Natural Science Foundation of China (42375089, W2411028), National Key Research and Development Program of China (2023YFC3709401 and 2023YFC3709400), and National Civilian Space Infrastructure Project (Grant No. Y5BZ31AC60).

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
