# Peer review of "Sea Breeze-Driven Daytime Vertical Distributions of Air Pollutants and Photochemical Implications in an Island Environment"

_EGUsphere, 2025_

## Author Comment (AC1)

**Response to Community Comments from Prof. Hai Guo**

We thank you for the constructive comments and suggestions, which are very positive to improve scientific contents of the manuscript. We have revised the manuscript appropriately and addressed your comments point-by-point for consideration as below. The remarks from yours are shown in black, our responses (in blue) and the corresponding edits in the manuscript (in red) are shown below. All the page and line numbers mentioned following are refer to the revised manuscript without change tracked.

General Comments

This study employs Multi-Axis Differential Optical Absorption Spectroscopy (MAX-DOAS) observations to characterize the vertical distributions of aerosols, $NO_2$, HCHO, and CHOCHO under different air current conditions (NSBDs, SBDs, and TDs) in a rural coastal area of Hainan Island, China. The results show that NSBDs are associated with higher pollutant levels and broader vertical distribution ranges. During SBDs, sea breezes limited pollutant dispersion, while their cooling effect suppressed $O_3$ formation. Under TDs, typhoons scavenge pollutants but facilitate mid-to-high altitude BVOC transport and enhance the vertical dispersion of surface pollutants. These findings provide valuable insights into coastal atmospheric processes and pollutant behaviors. The manuscript is generally well-structured and detailed. However, several issues need to be addressed before it can be considered for publication.

R: We sincerely thank you for the positive and constructive comments on our manuscript. The feedback is greatly appreciated and has been carefully considered in the revision. All concerns have been addressed to improve the clarity, rigor, and overall quality of the paper. We believe that the revised version has been significantly improved.

Specific Comments

1. Line 30: References should be consistently ordered, either alphabetically by author or chronologically.

R: Thank you for your comment. References have been arranged alphabetically by author's last name throughout the manuscript.

2. Line 85: Please clarify the time resolution of the MAX-DOAS observations.

R: Thank you for your comment. We have clarified that the temporal resolution of the MAX-DOAS measurements and please refer to Line 102-104.

"Data acquisition, with a temporal resolution of 30 seconds for individual spectrum, is conducted daily from 06:00 to 19:00 local time (LT), with dark current signals automatically extracted from nightly background measurements. A full scan sequence of different elevation angles takes approximately 5 minutes to complete."

3. Line 130: The sentences beginning with "Accurate identification of … of SB and LB" are redundant and should be revised.

R: Thank you for your comment. We have revised the sentences to remove redundancy. Please refer to Line 144-147.

4. Lines 140–150: I suggest moving Figure S4 from the Supplementary Material to the main text and citing it appropriately.

R: Thank you for your comment. Figure S4 has been supplemented into the main text as Figure 2 in Section 2.2. The manuscript has been revised accordingly, and all relevant citations have been updated.

5. Line 165: Please provide the formula used to calculate relative humidity. In addition, the ERA5 data may be too coarse to capture local weather conditions around the in-situ site. For example, uncertainties in planetary boundary layer height are significant. Verification against nearby weather station data is necessary to ensure the robustness of the meteorological analysis.

R: Thank you for your comment. We have presented the equations for RH conversion and relevant descriptions. Please refer to Line 185-192.

"Notably, ERA5 lacks the surface RH data required; therefore, we computed actual and saturated water vapor pressures using the 2m dew temperature ($T_d$ in K) and 2m temperature (T in K) from ERA5-Land, and derived RH from their ratio (Zhu et al., 2025):

$$RH = \frac{e_a(T_d)}{e_s(T)} \times 100\% \qquad (1)$$

where $e_a(T_d)$ is the actual vapor pressure at dew point temperature, and $e_s(T)$ is the saturation vapor pressure at the current temperature. These can be determined by the following formula:

$$e_a(T_d) = 0.6108 \times \exp\left(\frac{17.27 \times (T_d - 273.15)}{T_d - 35.86}\right) \qquad (2)$$

$$e_s(T) = 0.6108 \times \exp\left(\frac{17.27(T - 273.15)}{T - 35.86}\right) \qquad (3)"$$

Furthermore, we have conducted a comparative analysis of temperature and relative humidity (RH) between ERA5 and the nearest meteorological station to our study site—Danzhou station (19.31°N, 109.35°E, 170 m a.s.l.) (Fig. R1). However, it should be noted that although Danzhou station is the closest meteorological site available, it still far away from the measurement site (~33 km). Such distance may also contribute to the discrepancies in temperature and RH. Overall, the comparison shows relatively high agreement of temperature and RH between these two data sources.

[Figure]

*Figure R1. Correlation between Temperature (T_ERA5 and T_station) and Humidity (RH_ERA5 and RH_station) from ERA5_land and Danzhou Meteorological Station (19.31°N, 109.35°E) during summer 2024. The color of each hexagonal bin represents the frequency of data points within that grid cell. The yellow dashed line indicates the 1:1 line, and the green line shows the regression line between the two datasets.*

As shown in Fig. R2, the Danzhou station and ERA5_Land exhibit broadly consistent variations in WS and wind direction WD throughout the study period. The ERA5_Land WS is derived from the 10 m u- and v-wind components and represents grid-averaged conditions, whereas the Danzhou station is influenced by strong local surface roughness caused by nearby buildings and vegetation, resulting in lower observed WS. Despite these differences in magnitude, both datasets display similar prevailing wind directions on non-typhoon days, a shift toward southwesterly winds during typhoon periods, and comparable diurnal patterns in WS, as shown in Fig. R2(b), (c) and (d).

[Figure]

*Figure R2. Comparison of WS and WD between the Danzhou station and ERA5_Land. (a) Hourly variations in WS for the Danzhou station and ERA5_Land over the entire observation period; WD patterns at the (b) Danzhou station and (c) ERA5_Land during typhoon and non-typhoon days; (d) Diurnal variations of mean WS and WD for the Danzhou station and ERA5_Land on typhoon days. The gray shaded area indicates the range of the typhoon's duration. Wind direction is plotted in polar coordinates with percentage frequency indicated by concentric circles.*

Although ground-based observations can better capture near-surface meteorological conditions, their strong short-term fluctuations—particularly in wind fields—introduce substantial noise and complicate the identification of sea–land breeze circulations.

More importantly, the Danzhou station is located far from the measurement site, making its data unrepresentative to some extent. In contrast, ERA5_Land provides smoother and more continuous spatiotemporal fields that coherently depict the evolution of large- and mesoscale wind structures. As a result, ERA5_Land has been widely applied in sea–land breeze research worldwide and is generally regarded as reliable and internally consistent for such analyses (Azorin-Molina et al., 2011; Hallgren et al., 2023; Xia et al., 2022; Zhao et al., 2022).

In summary, despite certain uncertainties, ERA5_Land offers physically consistent estimates of all required variables and supplies complete, continuous, and quality-controlled time series—advantages that ground-based observations cannot fully match. The comparison with station measurements further confirms good agreement between these two datasets. These features make ERA5_Land particularly suitable for investigating sea–land breeze circulations in this study.

6. Line 175: Please provide the actual values instead of vague expressions such as "values recorded in Chinese megacity centers."

R: Thank you for your comment. We have provided average $NO_2$ concentration data for major Chinese megacity clusters, specifically reporting values of 7.62 ± 1.39 ppbv for the Beijing–Tianjin–Hebei and surrounding regions, and 7.45 ± 0.87 ppbv for the Yangtze River Delta. To ensure comparability between our observations and those from previous studies, we consulted additional literature containing concurrent records from the same monitoring period and re-compare the $NO_2$ levels in this study and those in the megacity clusters. The results indicate that the $NO_2$ concentration in this study was approximately one-third to one-fifth of that observed in the megacity clusters. Please refer to Line 205-208.

"The average $NO_2$ surface VMRs (1.61 ± 0.53 ppbv) was about 3 to 5 times lower than the values recorded in major Chinese megacity centers, such as the Beijing–Tianjin–Hebei region (7.62 ± 1.39 ppbv) and the Yangtze River Delta (7.45 ± 0.87 ppbv) (Lou et al., 2025; Ministry of Ecology and Environment of the People's Republic of China, 2024a, b, c), reflecting minimal local traffic and industrial emissions."

7. Figure S5: This figure is not cited in the text.

R: Thank you for your comment. We have to clarify that Figure S5 is intended to illustrate the average atmospheric conditions in the study region during the observational period. It is cited in Text S2, and used to determine the parameters required to accurately calculate the semidiurnal pressure waves and is not directly related to the main content of the paper.

8. Line 225: The statement of "Notably, most of the SBDs occur consecutively." can be explained by meso-scale circulations, which usually occur under stable atmospheric

conditions dominated by persistent high pressure.

R: Thank you for your comment. We have rephrased the sentence as following and please also refer to Line 267-268.

"Notably, most of the SBDs occur consecutively. This can be explained by the mesoscale circulation, which usually occur under stable atmospheric conditions dominated by persistent high pressure."

9. Line 245: The claim that "southwesterly winds persist under the typhoon's peripheral airflow" needs clarification. Since both the typhoon's location and the sampling site (in this and other studies) change over time, is the peripheral airflow always associated with southwesterly winds?

R: Thank you for your comment. Indeed, the typhoon's peripheral circulation does not always produce southwesterly winds at the study site; the WD depends on the typhoon's relative position and movement. For example, during the selected seven TDs, Figure R3 shows that when Typhoon "Prapiroon" approached Hainan on July 21, the peripheral circulation produced easterly to northerly winds. On July 22, as the typhoon moved northwestward and made landfall, the winds shifted to southwesterly. After its departure, southwesterlies dominated for several days due to the residual circulation. It illustrates that the peripheral airflow can vary substantially over time, and the resulting wind direction at a given site is determined by the typhoon's location, trajectory, and rotation rather than being inherently southwesterly.

We acknowledge that the original statement "southwesterly winds persist under the typhoon's peripheral airflow" may have been misleading. To avoid confusion, we have revised this sentence for greater precision. Please refer to Line 283-285.

"The prevailing southwesterly winds are caused by the typhoon's peripheral airflow over the area of measurement, whereas NSBDs feature daytime southwest winds shifting to southerly at night."

[Figure]

*Figure R3. Daily mean wind field characteristics of Hainan Island on seven consecutive typhoon days: (a) 2024-07-21, (b) 2024-07-22, (c) 2024-07-23, (d) 2024-07-24, (e) 2024-07-25, (f) 2024-07-26, and (g) 2024-07-27. Typhoon "Prapiroon" is indicated by a yellow spiral logo, and LT stands for local time.*

10. Line 260: Please add vertical wind components to the correlation figures to better support the conclusions.

R: Thank you for your comment. We have updated Fig. 5 with supplements of mean vertical velocity profiles for NSBDs, SBDs, and TDs, respectively. Please refer to the updated Fig. 5 (Fig. 6 in the revised manuscript and shown as Fig. R4 following) and related descriptions in Line 303-308. In addition, information on vertical velocity has been incorporated into Table S4 in the Supplementary Materials.

"As shown in Fig. 6(g) to (i), the differences in BLH further modulate vertical motion under the three ACPs. The positive values of vertical velocity ($\omega$) indicate downward airflow movement. Upward motion prevails below 2 km on NSBDs and TDs, associated with enhanced convection and typhoon-induced ascent, respectively. In contrast, SBDs feature weak morning uplift that transitions to subsidence as the sea breeze intensifies, with the strongest downward motion ($\omega \approx 0.05$ m s$^{-1}$) near 1.5 km, suggesting that the SB further restricts pollutant dispersion in FK."

[Figure]

*Figure R4. Diurnal variations of meteorological factors under different air currents. Mean diurnal variations of (a) WS, (b) WD, (c) temperature, (d) RH, (e) BLH, (f) UV radiation flux (UVF) and (g, h and i) vertical velocity (ω) under three air current patterns (ACPs): NSBDs, SBDs, and TDs. The gray shaded area represents the observation period of MAX-DOAS. The two black dotted lines define the range of WD for onshore and offshore winds. Critical transition points for SB initiation and cessation are marked with arrows in (b). The blue and brown arrows represent rising and sinking air currents, respectively.*

11. Section 3.3: It is unclear why $O_3$ concentrations are clustered instead of directly analyzing their evolution under NSBDs, SBDs, and TDs. Please clarify.

R: Thank you for your comment. It is indeed that the $O_3$ concentration analysis can be discussed either via their evolution under different ACPs or via clustered diurnal pattern. We know that NSBDs, SBDs and TDs are each associated with markedly different atmospheric conditions, including WS, RH, solar radiation, and vertical motion, which are also influencing $O_3$ diurnal pattern via both physical and chemical mechanisms. Besides, even within a same ACP, some of the meteorological parameters varied significantly. This variability makes it challenging to disentangle the relative contributions of individual factors, such as the effect of wind-driven transport versus photochemical production, or the roles of temperature, humidity, and vertical mixing in shaping the observed $O_3$ diurnal variations.

Figure R4 illustrates the diurnal $O_3$ profiles under NSBDs, SBDs, and TDs. The results show that although the absolute $O_3$ concentrations are substantially lower on SBDs

compared to NSBDs, the net increments of their diurnal variations are quite similar. This feature cannot be fully explained by weather-type classification alone, underscoring the added value of clustering analysis.

[Figure]

*Figure R5. Diurnal variations of O₃ surface VMR under different air currents. The net O₃ increment (ΔO₃) under each airflow pattern is displayed in the bar chart in the upper left corner.*

Clustering analysis takes a "bottom–up," data-driven approach, without imposing any a priori meteorological assumptions. Instead, it identifies inherent groups of similar O₃ behaviors directly. Then the influencing factors helping the enhancement of daily O₃ can be recognized from perspective on both ACPs and physical/chemical mechanisms. In general, integrating both approaches, i.e. O₃ diurnal patter classified with ACPs or clustered by itself, enable a more refined association between meteorological conditions and O₃ response, deepening our understanding of the drivers of O₃ pollution.

12. Lines 315–320: Is it reasonable to use the minimum early-morning O₃ concentration as background O₃? Have the effects of NO$_x$ titration been excluded?

R: Thank you for your comment. We fully agree that the early-morning O₃ concentration can indeed be influenced by NO$_x$ titration, and we acknowledge that defining it as background level is not so cautious. In this study, the minimum early-morning concentration was served as a baseline to calculate the daytime increment, which represents a net change. This metric of net change effectively allows for a clearer isolation and revelation of the net impact of the day's meteorological processes (i.e., sea breeze and non-sea breeze conditions) on local photochemical production efficiency. To avoid the misleading, we have corrected the sentence and please refer to Line 356-357.

"The differences between the midday peak and the morning minimum represents net O₃ increment (ΔO₃)."

13. The dataset includes only seven typhoon days, which may be insufficient to represent general typhoon-related airflows. This appears more like a case study.

R: Thank you for your comment. We acknowledge that the dataset of seven TDs may

appear limited from a purely statistical standpoint compared to the other two ACPs. However, Typhoon "Prapiroon" during observation period provided a valuable, albeit singular, opportunity to capture its direct influence on local air quality at measurement site. We have amended the manuscript to include a clarification regarding the case-study nature of this specific analysis and to avoid overgeneralization. Please refer to Line 260-266.

"During the campaign, a unique opportunity arose to study the impact of Typhoon "Prapiroon", which traversed Hainan from southeast to northwest on July 21–23, 2024. Its intense disturbances to the local meteorology continued for one week after landfall (Fig. 1a and 6). Although the seven typhoon-influenced days (TDs) may be not sufficient sample from a statistical perspective, they represent a coherent case of a recurring meteorological phenomenon—typhoons following a characteristic east-to-west track across the South China Sea. This consistent pathway results in systematic advection of specific air masses into the study region. So we treated these TDs as an independent case against NSBDs and SBDs identified by the ORA (Table S3)."

We also wish to emphasize the strong meteorological rationale for considering these events together. Typhoons originating in the South China Sea and moving past Hainan Island almost invariably follow a consistent east-to-west trajectory. This predictable pathway produces a systematic transport mechanism, consistently advecting distinct air masses from the typhoon's trajectory into our study region. Therefore, while the seven-day sample size is modest, it represents observations of a coherent and recurring meteorological phenomenon rather than seven random and disparate events. This consistency strengthens the relevance of our findings for understanding the air quality impacts of typhoons taking this common track.

14. Please clarify whether the results apply only to Hainan Island or can be generalized to other coastal/island environments. Which findings can be extended to similar regions?

R: Thank you for your comment. We note that a similar comment was raised by another reviewer (Reviewer #1, Comment 6), so we have added a dedicated paragraph in the conclusion section outlining the extent to which our findings can be generalized and also the key limitations of our study. Please refer to Line 535-543.

"This study reveals several summertime physical and chemical processes that are characteristic of Hainan Island, so the quantitative values reported here—such as pollutant concentrations, ORA parameter ranges and FNR/GNR thresholds—are shaped by the island's unique land–sea configuration, monsoon regime, and emission environment and should be treated as site-specific. However, the vertical stratification mechanisms linked to species lifetimes and atmospheric dynamics, the thermodynamic and compositional effects of SB intrusions, the typhoon-related scavenging, redistribution, and uplift, and the diagnostic value of different photochemical indicators under different ACPs are processes that commonly occur across many coastal and island settings. Thus, while specific numerical thresholds may vary, the underlying conceptual processes and the ORA framework offer a transferable basis for interpreting

similar coastal phenomena, particularly in regions with comparable climatic and geographic conditions."

Technical Corrections

1. Ensure consistent use of abbreviations: introduce the full term at first mention, followed by the abbreviation, and use the abbreviation thereafter.

R: Thank you for your comment. We have carefully checked and revised the manuscript to ensure that all abbreviations are introduced with their full terms at the first mention, followed by the abbreviation in parentheses. Thereafter, only the abbreviations are used consistently throughout the text.

2. Line 60: Add a comma before "Nevertheless."

R: Thank you for your comment. A comma has been added and please refer to Line 72.

3. Line 135: Use "Fig. 1a and b" instead of "Fig. 1a,b."

R: Thank you for your comment and we have corrected accordingly. Please refer to Line 148.

4. Correct typos, grammatical errors, and syntactic mistakes throughout the manuscript.

R: Thank you for your comment. We have thoroughly checked the manuscript and corrected all typos, grammatical errors, and syntactic mistakes to improve clarity and readability.

5. The English should be polished further, ideally by a native speaker.

R: Thank you for your comment. we have polished the manuscript carefully, corrected grammatical mistakes and expressed more academically and briefly.

**Reference:**

Azorin-Molina, C., Tijm, S., and Chen, D.: Development of selection algorithms and databases for sea breeze studies, Theor. Appl. Climatol., 106, 531-546, https://doi.org/10.1007/s00704-011-0454-4, 2011.

Hallgren, C., Körnich, H., Ivanell, S., and Sahlée, E.: A Single-Column Method to Identify Sea and Land Breezes in Mesoscale-Resolving NWP Models, Weather Forecasting, 38, 1025-1039, https://doi.org/10.1175/WAF-D-22-0163.1, 2023.

Lou, Y., Chen, Y., Chen, X., and Li, R.: Spatiotemporal estimates of anthropogenic $NO_x$ emissions across China during 2015-2022 using a deep learning model, J. Hazard. Mater., 487,

https://doi.org/10.1016/j.jhazmat.2025.137308, 2025.

Ministry of Ecology and Environment of the People's Republic of China: National Urban Air Quality Report for August 2024, https://www.mee.gov.cn/hjzl/dqhj/cskqzlzkyb/202409/W020240924585303726990.pdf, 2024a.

Ministry of Ecology and Environment of the People's Republic of China: National Urban Air Quality Report for July 2024, https://www.mee.gov.cn/hjzl/dqhj/cskqzlzkyb/202408/W020240823817374806303.pdf, 2024b.

Ministry of Ecology and Environment of the People's Republic of China: National Urban Air Quality Report for June 2024, https://www.mee.gov.cn/hjzl/dqhj/cskqzlzkyb/202407/W020240725571411086492.pdf, 2024c.

Xia, G., Draxl, C., Optis, M., and Redfern, S.: Detecting and characterizing simulated sea breezes over the US northeastern coast with implications for offshore wind energy, Wind Energ. Sci., 7, 815-829, https://doi.org/10.5194/wes-7-815-2022, 2022.

Zhao, D., Xin, J., Wang, W., Jia, D., Wang, Z., Xiao, H., Liu, C., Zhou, J., Tong, L., Ma, Y., Wen, T., Wu, F., and Wang, L.: Effects of the sea-land breeze on coastal ozone pollution in the Yangtze River Delta, China, Sci. Total Environ., 807, https://doi.org/10.1016/j.scitotenv.2021.150306, 2022.

Zhu, M., Zhu, D., Huang, M., Gong, D., Li, S., Xia, Y., Lin, H., and Altan, O.: Assessing the Impact of Climate Change on the Landscape Stability in the Mediterranean World Heritage Site Based on Multi-Sourced Remote Sensing Data: A Case Study of the Causses and Cévennes, France, Remote Sens., 17, 203, https://www.mdpi.com/2072-4292/17/2/203, 2025.

---

## Author Comment (AC2)

**Response to Comments from Reviewer #2**

We thank you for the constructive comments and suggestions, which are very positive to improve scientific contents of the manuscript. We have revised the manuscript appropriately and addressed your comments point-by-point for consideration as below. The remarks from yours are shown in black, our responses (in blue) and the corresponding edits in the manuscript (in red) are shown below. All the page and line numbers mentioned following are refer to the revised manuscript without change tracked.

Reviewer #2: Li et al. combine MAX-DOAS observations with an objective sea–land breeze identification algorithm to characterize pollutants, e.g. $NO_2$, HCHO, CHOCHO, spatiotemporal distributions and photochemical indicators for an island environment under distinct airflow regimes, including sea breezes and typhoons. The results prove the critical role of local meteorology in modulating pollution levels, vertical profiles, and further photochemical indicators (FNR and GNR), which are influencing the ozone formation sensitivity across different altitudes. These findings provide valuable insights into the interplay between meteorological dynamics and atmospheric chemistry in tropical coastal environments. Overall, the manuscript is logically organized, clearly illustrated, and well-written. I recommend its acceptance after minor revisions addressing the following points.

R: We would like to express our sincere gratitude for your careful review and for the insightful and constructive comments provided. We also greatly appreciate your positive and encouraging remarks regarding the overall quality and presentation of our work. Your feedbacks are crucial in helping us improve the clarity and coherence of the manuscript. We have carefully addressed all the suggestions and made corresponding revisions, which we believe have further strengthened the paper.

Specific comments:

1. P1, L15: Hainan→Hainan, China

R: Thank you for your comment. We have revised the text to read "Hainan, China" as suggested and please refer to Line 15.

2. P1, L22: "existing OFS classifications" is unclear. Does it mean the classification methodology or the thresholds?

R: Thank you for your comment. We recognize that the original wording could be ambiguous, and we have revised the text to clarify that "existing OFS classifications" refers specifically to the classification thresholds. Please refer to Line 22-24.

"Elevated FNR and GNR thresholds suggest that existing OFS classification thresholds are inadequate for low-$NO_2$ tropical coastal rural areas, underscoring the need for

region-specific assessments."

3. P1, Abstract: it could briefly clarify the observational period or duration (e.g., season or year), which would help readers contextualize the results temporally.

R: Thank you for your comment. We have added the observational period in the Abstract to better contextualize the study temporally. Please refer to Line 13-16.

"Utilizing Multi-Axis Differential Optical Absorption Spectroscopy (MAX-DOAS) and a sea-land breeze objective identification algorithm, we reveal distinct spatiotemporal patterns in $NO_2$, HCHO, CHOCHO, and associated photochemistry under varying airflow patterns in a rural coastal area of Hainan, China during the summer of 2024."

4. P2, L37-38: As described, the MAX-DOAS method has been successfully applied for atmospheric compositions monitoring. How about the performances especially for $NO_2$, HCHO and CHOCHO?

R: Thank you for your comment. We have added a short discussion on the performances of MAX-DOAS retrievals for $NO_2$, HCHO, and CHOCHO in the revised manuscript. Specifically, as a mature optical remote sensing technique, MAX-DOAS has been widely applied for monitoring tropospheric trace gases. This method provides high sensitivity and accuracy in detecting atmospheric constituents and enables the retrieval of vertical distribution profiles, making it a powerful tool in studies of air pollution mechanisms, model and satellite validation, and source characterization (B. Ren et al., 2021; H. Ren et al., 2021; Tan et al., 2018).

For $NO_2$ and HCHO, the MAX-DOAS technique has established well-validated retrieval frameworks. Numerous studies have demonstrated that MAX-DOAS–derived vertical column densities and surface concentrations show good agreements with those from satellite observations and in situ measurements (Chan et al., 2020; Jiang et al., 2025; Kumar et al., 2020; Martin et al., 2004; Schreier et al., 2020; Wagner et al., 2011). Furthermore, MAX-DOAS has proven capable of providing stable long-term observations of $NO_2$ and HCHO even under complex atmospheric conditions such as mountainous regions, open oceans, and typhoon environments (Schreier et al., 2016; Zhang et al., 2022).

For CHOCHO, whose concentrations in the atmosphere are relatively low and difficult to monitor by conventional methods, MAX-DOAS remains one of the few ground-based techniques capable of reliable detection. Despite challenges from noise and spectral interferences, successful retrievals of CHOCHO have been reported in both urban and forest environments, yielding valuable insights into its photochemical behavior and its ratio to HCHO (Hong et al., 2024; Hoque et al., 2018; Liu et al., 2021; Schreier et al., 2020; Zhang et al., 2025). In addition, CHOCHO measurements from MAX-DOAS have been effectively used to validate satellite products such as TROPOMI and OMI (Lerot et al., 2021).

In summary, MAX-DOAS demonstrates robust and reliable performance in monitoring $NO_2$, HCHO, and CHOCHO, providing strong methodological support for the data quality in this study. Please refer to Line 40-48.

"Previous studies reported that well-established retrieval frameworks for $NO_2$ and HCHO have shown good agreement with satellite and in situ measurements, demonstrating its robustness for long-term and multi-environment observations (Chan et al., 2020; Jiang et al., 2025; Kumar et al., 2020; Martin et al., 2004; Schreier et al., 2020; Schreier et al., 2016; Wagner et al., 2011; Zhang et al., 2022). CHOCHO is challenging to detect due to its weak absorption and spectral interferences, yet MAX-DOAS remains one of the few ground-based techniques capable of reliable retrievals. Successful measurements in both urban and biogenic regions have advanced understanding of CHOCHO photochemistry, which also were validate satellite products (Hong et al., 2024; Hoque et al., 2018; Lerot et al., 2021; Liu et al., 2021; Schreier et al., 2020; Zhang et al., 2025). Overall, MAX-DOAS demonstrates robust and reliable performance in monitoring $NO_2$, HCHO, and CHOCHO, providing strong methodological support for the data quality in this study."

5. P2, L44, "elevated $R_{GF}$ at higher altitudes" means even more glyoxal and less formaldehyde, what is the relationship between this ratio variation and anthropogenic VOCs?

R: Thank you for your comment. Indeed, the $R_{GF}$ is used to identify the dominance of BVOCs or AVOCs emission (Chan Miller et al., 2014; Chen et al., 2023; DiGangi et al., 2012; Kaiser et al., 2015; MacDonald et al., 2012; Vrekoussis et al., 2010; Xing et al., 2020). It reported that higher $R_{GF}$ values are associated with anthropogenic sources; for example, _ENREF_17Vrekoussis et al. (2010) report $R_{GF} < 0.04$ in isoprene-dominated areas and substantially higher values where biomass burning, monoterpene emissions, or anthropogenic emissions prevail. This threshold has been supported by subsequent studies (Chen et al., 2023). Moreover, numerous studies have reported a gradual increase in the $R_{GF}$ with altitude within the 0-3 km layer (Baidar et al., 2013; Hong et al., 2022; Zhang et al., 2025), with values in the free troposphere even exceeding those in the boundary layer (Kaiser et al., 2015).

6. P3, L57: distribution? Or formation?

R: Thank you for your comment. Upon careful review, we found that the original term "distribution" was used incorrectly here. The intended meaning is "formation," and this has been corrected in the revised manuscript. Please refer to Line 67.

7. P3, L62: Please add a full stop between "layers" and "Nevertheless"

R: Thank you for your comment. We have added a full stop between "layers" and "Nevertheless" in the revised manuscript. Please refer to Line 72.

8. P4, L83: altitude …. above sea level

R: Thank you for your comment. We have revised the sentence to clarify the altitude reference as "above sea level." Please refer to Line 97.

9. P4, L94-95: What is RMS? And why did higher error thresholds set for HCHO and CHOCHO?

R: Thank you for your comment. RMS refers to the root-mean-square residual between the measured and fitted differential optical absorption spectra, which reflects the quality of spectral fitting. We have added the full form of the abbreviation for clarity and please refer to Line 110-111.

"Quality control excluded retrievals with Root Mean Square (RMS) > 0.001 and DSCD error to DSCD ratios exceeding 10% (30% for HCHO, 50% for CHOCHO)."

Higher DSCD error to DSCD ratios thresholds were set for HCHO and CHOCHO because their absorption structures are relatively weaker and more susceptible to spectral interferences compared with aerosol and $NO_2$ (Jiang et al., 2025; Schreier et al., 2020). Therefore, a slightly relaxed threshold ensures sufficient data coverage while maintaining acceptable retrieval accuracy.

10. P5, L110: Regarding the surface concentration of the a priori profile, is it reasonable to set HCHO even higher than as twice as $NO_2$ and twenty as HCHO? How the surface concentration of the a priori profile influencing the profile retrieval?

R: Thank you for your comment. The prior surface concentrations (SC) of $NO_2$, HCHO, and CHOCHO (2, 5, and 0.2 ppbv) were selected based on typical background conditions in the study region. A foreseeable feature of the study region is that its rural setting leads to relatively low $NO_2$ levels, whereas strong biogenic emissions compensate for this and result in HCHO and CHOCHO concentrations comparable to those reported in some urban areas. CHOCHO is generally about one-twentieth of HCHO, with previous studies even setting it as low as one-sixtieth (Xing et al., 2020; Zhang et al., 2025). Therefore, the SC selected by this study is scientifically sound and supported by ample literature evidence.

To assess the impact of the prior SC, we first surveyed a wide range of published MAX-DOAS studies and summarized the commonly used values: typical priors for $NO_2$ include 1, 2, 5, and 10 ppbv (Hong et al., 2024; Jiang et al., 2025; Xing et al., 2022; Xing et al., 2020; Zhang et al., 2021); for HCHO, 1, 3, and 5 ppbv are frequently adopted and CHOCHO commonly used values are 0.05, 0.1, 0.2, and 0.5 ppbv (Hong et al., 2024; Xing et al., 2022; Xing et al., 2020; Zhang et al., 2025). We performed sensitivity texts across these values (Fig. R1). The results show that higher priors increase the retrieved near-surface values, but the influence gradually stabilizes. Vertical profiles of HCHO and CHOCHO become increasingly similar as they approach

realistic ambient concentrations. CHOCHO, however, exhibits substantially larger variability: its ambient concentrations are low and its absorption features are weak, so measurement noise—especially at higher altitudes—has a non-negligible impact on the retrieved profiles. NO₂, owing to its stronger absorption features and higher sensitivity near the surface, provides tighter observational constraints. As a result, increases in the SC lead to only modest changes aloft.

[Figure]

*Figure R1. Example of profile retrievals for (a) NO₂, (b) HCHO, and (c) CHOCHO on 15 June 2024. The a priori surface concentrations (SC) were set to 2, 5, and 0.2 ppbv as the baseline values, and additional sensitivity tests were conducted using alternative priors within the ranges commonly reported in literature: NO₂ (1, 2, 5, 10 ppbv), HCHO (1, 3, 5 ppbv), and CHOCHO (0.05, 0.1, 0.2, 0.5 ppbv). The corresponding retrieved surface concentrations (averaged over 0–100 m) are shown in the upper-right panels.*

The differences in vertical profile shapes under varying prior SC mainly stem from the structure of the optimal estimation framework. When the prior is low, the upper parts of the HCHO and CHOCHO profiles are dominated by the prior because MAX-DOAS sensitivity decreases rapidly with altitude. In this case, the inversion reduces the cost function by increasing concentrations in the highly sensitive lower layers while keeping the upper layers close to the prior. For example, with particularly low priors for HCHO (SC=1 ppb) and CHOCHO (SC=0.05 ppb), the observational constraint aloft becomes very weak, causing the upper layers to revert almost entirely to the prior, whereas the lower layers are substantially elevated by the measurements. This characteristic pattern and the underlying OEM behavior have been extensively discussed in previous studies (Frieß et al., 2019; Vlemmix et al., 2015; Wang et al., 2013).

Overall, the setting of the prior profile fully accounts for the typical atmosphere background and source characteristics of the study region, as well as the known altitude-dependent sensitivity of MAX-DOAS retrievals. The selected prior values fall well within the ranges reported in previous studies and behave consistently with the expected optimal estimation response. Therefore, the a priori settings used in this work are scientifically sound and reliable.

11. P5, L115: The unit for ozone should be corrected.

R: Thank you for your comment. We have corrected the unit for ozone to the appropriate one in the revised manuscript. Please refer to Line 131.

12. Fig 1: the label for typhoon track should be added in Fig. 1a.

R: Thank you for your comment. We have added the label for the typhoon track in Fig. 1a in the revised manuscript.

13. P6, Line127-132: There are two sentences with a certain degree of repeatability.

R: Thank you for your comment. We have revised the sentences to remove redundancy.

14. P7, Line 140: Please supplement a wind rose plot for the campaign, better with a separate daytime and nighttime one.

R: Thank you for your comment. We have plotted the wind rose for daytime and nighttime during the observation period (Fig. R2) and included in the Supplementary Material as Fig. S4. The description in the manuscript has also been updated and please refer to Line 154-156.

"Notably, during summer, the island is predominantly influenced by the southwesterly monsoon, which frequently produces daytime background winds oriented landward or along the coast (Fig. S4)."

In supplementary materials:

[Figure]

*Figure R2. Wind rose plots during the observation period for daytime (a) and night (b). The gray shaded area indicates upshore WD, and the gray dashed line represents the coastline. WD is plotted in polar coordinates with percentage frequency indicated by concentric circles.*

15. P7, Sect. 2.2 and 2.3: Regarding the identification of SBL, there are two questions: 1) how to evaluate the identification performance? Can it be validated? 2) how about the representativeness of the meteorological data used in the identification? Would it cause some uncertainties?

R: Thank you for your comment. Regarding the first question, the filtering method employed in this study is a well-established approach for identifying SBDs. To evaluate its performance, we randomly selected four days identified by the algorithm as SB events and analyzed the diurnal variations in surface WS, WD, temperature, RH, total

precipitation (TP), and surface pressure (SP) (Fig. R3). The results clearly show that all cases exhibited typical daytime sea breeze penetration and nighttime land breeze development, characterized by distinct wind shifts accompanied by either temperature drops or RH increases upon the arrival of the SB.

[Figure]

*Figure R3. Time series of WS, WD, temperature, RH, Total precipitation (TP) and surface pressure (SP)throughout the four randomly selected case days. The gray shaded areas indicate periods of onshore flow.*

Furthermore, the surface wind field confirms that the study area experienced weak background wind conditions during these periods, providing a favorable environment for the development of sea breeze systems (Fig. R4). Specifically, beginning around 09:00 LT, the wind field in the study area initiated a gradual shift, evolving into a stable onshore flow by 14:00 LT. This clear progression, consistent with the established physical mechanism of SB development, collectively validates the effectiveness of ORA.

[Figure]

*Figure R4. Surface wind fields for the four randomly selected cases, depicting the evolution at 04:00 LT, 09:00 LT, 14:00 LT, and 18:00 LT. The wind vector scaling is annotated in the lower right panel.*

Although previous studies have reported sea-breeze frequencies over Hainan Island, the absence of published event dates prevents direct comparison (Liang and Wang, 2017; Liang et al., 2017; Tang, 2015; Zhang et al., 2014). Given the short study period, we were able to diagnose each day using an established manual identification method (MIM), which has been applied and validated in earlier work (Azorin-Molina et al., 2011). Following this method, we identified 34 SBDs and compared them day by day with the ORA results, with the outcomes summarized in a confusion matrix (Table R1). Here, NSBDs refer to all remaining days after removing the manually identified SBDs.

Based on the cross matrix, the following standard performance metrics were calculated: (a) Overall Accuracy (ACC), representing the proportion of days on which both methods agree; (b) Probability of Detection (POD), representing the proportion of SBDs identified by the MIM that were also successfully detected by the ORA; (c) False Alarm Ratio (FAR), representing the proportion of SBDs identified by the ORA that were not confirmed by the MIM; and (d) F1-Score, defined as the harmonic mean of POD and Precision (1 − FAR), which provides a comprehensive measure of model performance. The F1-Score ranges from 0 to 1, with higher values indicating better performance.

The calculated values were: ACC = 90.4%, POD = 82.4%, FAR = 9.7%, and an F1-Score = 86.2%. These results indicate strong agreement between our filtering algorithm

and the manual reference. The high overall accuracy and F1-score confirm the algorithm's robust performance. The primary discrepancy, reflected in the slightly lower POD, is attributable to the objective algorithm's stricter criterion for the persistence of onshore flow, particularly on days with less pronounced wind shifts (For details, see Channel 1 in the Text S1 and Fig. 2).

*Table R1. Cross Matrix: ORA versus MIM.*

| Category | MIM: SBDs | MIM: NSBDs |
|---|---|---|
| ORA: SBDs | True Positive days ($D_{TP}$): 28 days | False Positive days ($D_{FP}$): 3 days |
| | Both methods agree are SBDs | Only ORA identifies as SBDs |
| ORA: NSBDs | False Negative days ($D_{FN}$): 6 days | True Negative days ($D_{TN}$): 57 days |
| | Only MIM identifies as SBDs | Both methods agree are NSBDs |

Regarding the second question, we conducted a comparative analysis of temperature and relative humidity (RH) between ERA5_Land and the nearest meteorological station to our study site—Danzhou station (19.31°N, 109.35°E, 170 m a.s.l.) (Fig. R5). However, it should be noted that although Danzhou station is the closest meteorological site available, it still far away from the measurement site (~33 km). Such distance may also contribute to the discrepancies in temperature and humidity. Overall, the comparison shows relatively high agreement of temperature and RH between these two data sources.

[Figure]

*Figure R5. Correlation between Temperature (T_ERA5 and T_station) and Humidity (RH_ERA5 and RH_station) from ERA5_land and Danzhou Meteorological Station (19.31°N, 109.35°E) during summer 2024. The color of each hexagonal bin represents the frequency of data points within that grid cell. The yellow dashed line indicates the 1:1 line, and the green line shows the regression line between the two datasets.*

As shown in Fig. R6, the Danzhou station and ERA5_Land exhibit broadly consistent variations in WS and wind direction WD throughout the study period. The ERA5_Land WS is derived from the 10 m u- and v-wind components and represents grid-averaged conditions, whereas the Danzhou station is influenced by strong local surface roughness caused by nearby buildings and vegetation, resulting in lower observed WS. Despite these differences in magnitude, both datasets display similar prevailing wind directions on non-typhoon days, a shift toward southwesterly winds during typhoon periods, and

comparable diurnal patterns in WS, as shown in Fig. R6(b), (c) and (d).

[Figure]

*Figure R6. Comparison of WS and WD between the Danzhou station and ERA5_Land. (a) Hourly variations in WS for the Danzhou station and ERA5_Land over the entire observation period; WD patterns at the (b) Danzhou station and (c) ERA5_Land during typhoon and non-typhoon days; (d) Diurnal variations of mean WS and WD for the Danzhou station and ERA5_Land on typhoon days. The gray shaded area indicates the range of the typhoon's duration. Wind direction is plotted in polar coordinates with percentage frequency indicated by concentric circles.*

Although ground-based observations can better capture near-surface meteorological conditions, their strong short-term fluctuations—particularly in wind fields—introduce substantial noise and complicate the identification of sea–land breeze circulations. More importantly, the Danzhou station is located far from the measurement site, making its data unrepresentative. In contrast, ERA5_Land provides smoother and more continuous spatiotemporal fields that coherently depict the evolution of large- and mesoscale wind structures. As a result, ERA5_Land has been widely applied in sea–land breeze research worldwide and is generally regarded as reliable and internally consistent for such analyses (Azorin-Molina et al., 2011; Hallgren et al., 2023; Xia et al., 2022; Zhao et al., 2022).

In summary, despite certain uncertainties, ERA5_Land offers physically consistent estimates of all required variables and supplies complete, continuous, and quality-controlled time series—advantages that ground-based observations cannot fully match. The comparison with station measurements further confirms good agreement between these two datasets. These features make ERA5_Land particularly suitable for investigating sea–land breeze circulations in this study.

16. P8, Line 170, daily averaged

R: Thank you for your comment. The missing word "averaged" after "daily" has been added in the revised manuscript. Please refer to Line 198.

17. P8, Line 173-175, Even in Beijing and Shanghai, AOD should also be lower in summer, how to get the conclusion of "typically exceeds 0.5"?

R: Thank you for your comment. Upon reviewing the latest literature, we found that the previously cited study on Beijing covers a period considerably earlier than this study, and its conclusions are therefore no longer directly applicable. After reviewing the recent reported averaged summer AOD in Beijing and Yangtze River Delta area, we have revised the manuscript accordingly to ensure that the discussion reflects the most up-to-date observations. Please refer to Line 201-203.

"The observed summer aerosol loading (mean AEC and AOD measuring $0.11 \pm 0.03$ km$^{-1}$ and $0.29 \pm 0.1$, respectively) exhibited markedly lower values compared to urban agglomerations like Beijing and Shanghai (where AOD is around 0.4) (Fan et al., 2025; Peng et al., 2025)……"

18. P9, Line 186: how to quantitatively evaluate "a clear decoupling between surface and column measurements for NO$_2$, HCHO, and CHOCHO, which was more pronounced in aerosols"?

R: Thank you for your comment. To quantitatively evaluating, we have analyzed relationships between surface and column integrated concentrations of NO$_2$, HCHO, and CHOCHO, as well as between surface aerosol extinction coefficients (AEC) and aerosol optical depth (AOD). The degree of decoupling can be inferred from the dispersion of data points—greater deviations from the linear regression line indicate weaker coupling between surface and column values. As shown in Fig. R7, both the $R^2$ (0.34) and Pearson correlation coefficient (0.58) between AEC and AOD are notably lower than those for the trace gases, confirming a more pronounced decoupling in aerosol distributions. We have supplemented the relevant correlation results and discussion. Please refer to Line 221-223 and Fig. S7.

"…, which was more pronounced in aerosols as indicated by the weakest correlation ($R^2$=0.34) between surface AEC and AOD (Fig. S6). These results suggest that the vertical distribution pattern of pollutants may involve more complex physicochemical processes that deserve further investigation."

[Figure]

*Figure R7. Correlations between surface and column integrated concentrations of (a) aerosol; (b) NO₂; (c) HCHO; and (d) CHOCHO during the observation period.  The black dashed line represents the scaled reference line, while the blue solid line indicates the linear regression line. The regression equation, coefficient of determination (R²), and Pearson correlation coefficient are shown in the upper-left corner of each panel.*

19. P10, Line 199-202: Similarly, could the authors provide any quantitatively evidence between humidity and extinction to certificate the aerosol hygroscopic growth?

R: Thank you for your comment. In the Supplementary Materials, Figure R8 (as Figure S8) shows the Pearson correlation coefficients (r) between AEC and RH at each altitude during the same time. The results indicate that the r coefficients between aerosols and RH at the same altitude are generally above 0.6. Particularly at an altitude of approximately 1 km, the correlation coefficient reaches 0.86, indicating a significant positive correlation between the two variables. Notably, negative correlations are observed in the lower boundary layer (0.3 and 0.5 km), which likely reflect the complex interplay between boundary-layer dynamics, local emissions, and turbulent mixing: near-surface aerosol concentrations can be influenced by local emissions that do not coincide with humidity peaks, while diurnal boundary-layer development and turbulent transport can redistribute aerosols, temporarily decoupling their variability from RH. We have supplemented the corresponding section of the manuscript with quantitative results to strengthen the robustness of these findings. Please refer to Line 232-236.

"Notably, the aerosol vertical distribution bears a remarkable similarity to the RH. The Pearson correlation between AEC and RH remains positive above 0.5 km, reaching a maximum of 0.86 around 1 km, indicating a strong positive relationship at this altitude (Fig. S8). This phenomenon reflects aerosol hygroscopic growth characteristics, particularly the water vapor sensitivity of fine particulate matter, which enhances light extinction at these altitudes (Liu et al., 2020)."

[Figure]

*Figure R8. Pearson correlation coefficients (r) between AEC and RH at each altitude during the same time.*

20. P14-16: The discussion is too informative. Better to give a brief summary to characterize the difference of NO₂、HCHO and CHOCHO between NSBDs and SBDs?

And how these differences regulated by the ACPs.

R: Thank you for your comment. We have revised the discussion to provide a more concise summary. Now, instead of a detailed descriptive account, we highlight the key differences in $NO_2$, HCHO and CHOCHO between NSBDs and SBDs, and briefly explain how these differences are influenced by the ACPs. This makes the comparison clearer and keeps the focus on the mechanisms rather than extensive descriptive details. Please refer to Page 17-19.

21. Fig. S11: net ozone production ($\Delta O_3$) → net ozone increment

R: Thank you for your comment. We have revised the term "net ozone production ($\Delta O_3$)" to "net ozone increment" in the caption for Fig. S11 (now Figure S14).

22. P21, Line 390-401: Can be draw a conclusion that which indicator, i.e. GNR and FNR, is better for diagnose the ozone formation sensitivity?

R: Thank you for your comment. We have discussed this issue in detail on Line 474-494. In brief, our $R_{GF}$ analysis indicates that sea-breeze and typhoon events transport BVOCs, which tends to shift the ozone chemical regime toward $NO_x$-limited or transition conditions. The GNR results are consistent with this shift, whereas FNR shows a weaker response in the 400–1200 m layer, particularly under sea-breeze and typhoon conditions (Fig. 11). This suggests that GNR more reliably reflects changes in OFS. We note that FNR can be influenced by primary HCHO sources, which may bias sensitivity diagnostics (Liu et al., 2021), but our study lacks quantitative constraints to separate these effects. Overall, under BVOC-enhanced conditions, GNR provides a more robust indication of the shift toward $NO_x$-limited or transition regimes than FNR, though further investigation is needed for full validation. We also followed your suggestion by removing unnecessary details and replacing them with more concise, summary-level descriptions to make the text clearer and more focused. Please refer to Line 481-494.

23. Fig. 10: It can be seen that the results of ozone formation sensitivity are quite different obtained by using different indicators. So which one is more reliable? Or any preference for using under different ACPs?

R: Thank you for your comment. This issue has been addressed in our response to Comment #22. As discussed, GNR performs more consistently under BVOC-enhanced or BVOC-dominated conditions, whereas FNR may be biased by primary HCHO emissions. Further validation is still needed, as detailed in our response to Comment 22 and in the revised manuscript. Please refer to Line 474-494.

24. Also the OFS regimes shifted across different layers, can it be driven by some key meteorological parameters in view of the ACPs?

R: Thank you for your comment. The vertical distribution of the OFS is closely linked to the relative concentrations of $NO_2$, HCHO, and CHOCHO, which are in turn strongly influenced by meteorological conditions. Among these factors, wind direction plays the dominant role.

For example, during the passage of Typhoon "Prapiroon", although the typhoon's overall trajectory moved from southeast to northwest across the study region, the near-surface winds remained predominantly southwesterly, while winds above 1 km shifted to westerly (Fig. R9 and also Fig. S12). On the one hand, the typhoon's movement transported BVOCs emitted from the Wuzhi Mountain toward the measurement site. On the other hand, the local wind structure facilitated the inflow of pollutants from upwind regions with high $NO_2$ emissions. Notably, at higher altitudes, the WD was consistent with that of the near-surface sea breeze, favoring the presence of $NO_2$ aloft. In addition, persistent upward motion likely enhanced this vertical pattern, leading to VOC-limited region in the upper layers as indicated by both FNG and GNR diagnostics. In contrast, other meteorological parameters—such as temperature, RH, and BLH— exerted secondary influences in comparison.

[Figure]

*Figure R9. Wind profiles for different air current patterns. The lengths of the arrows indicate the magnitude of the wind speeds and they are subjected to the same scaling factor on the three air current patterns (ACPs). The gray dotted line refers to the $O_{time}$ of the SB.*

We have also made a brief summary about this issue in Line 484-486.

"An interesting feature is that both metrics exhibit VOC-limitation aloft on TDs, likely because upper-level winds align with the SB pathway and transport upstream $NO_x$ into the region. Such behavior shows that WD may be a key factor driving OFS transitions across different ACPs (Fig. S12)."

25. A general comment that it is too informatively described in the manuscript. I recommend the authors can simplify and shorten some parts to some degree.

R: Thank you for your comment. In the revised manuscript, we have streamlined several sections by removing redundant descriptions and condensing overly detailed explanations. These revisions help to improve readability and highlight the key findings more clearly.

**Reference:**

Azorin-Molina, C., Tijm, S., and Chen, D.: Development of selection algorithms and databases for sea

breeze studies, Theor. Appl. Climatol., 106, 531-546, https://doi.org/10.1007/s00704-011-0454-4, 2011.

Baidar, S., Oetjen, H., Coburn, S., Dix, B., Ortega, I., Sinreich, R., and Volkamer, R.: The CU Airborne MAX-DOAS instrument: vertical profiling of aerosol extinction and trace gases, Atmos. Meas. Tech., 6, 719-739, https://doi.org/10.5194/amt-6-719-2013, 2013.

Chan, K. L., Wiegner, M., van Geffen, J., De Smedt, I., Alberti, C., Cheng, Z., Ye, S., and Wenig, M.: MAX-DOAS measurements of tropospheric $NO_2$ and HCHO in Munich and the comparison to OMI and TROPOMI satellite observations, Atmos. Meas. Tech., 13, 4499-4520, https://doi.org/10.5194/amt-13-4499-2020, 2020.

Chan Miller, C., Gonzalez Abad, G., Wang, H., Liu, X., Kurosu, T., Jacob, D. J., and Chance, K.: Glyoxal retrieval from the Ozone Monitoring Instrument, Atmos. Meas. Tech., 7, 3891-3907, https://doi.org/10.5194/amt-7-3891-2014, 2014.

Chen, Y., Liu, C., Su, W., Hu, Q., Zhang, C., Liu, H., and Yin, H.: Identification of volatile organic compound emissions from anthropogenic and biogenic sources based on satellite observation of formaldehyde and glyoxal, Sci. Total Environ., 859, 159997, https://doi.org/10.1016/j.scitotenv.2022.159997, 2023.

DiGangi, J. P., Henry, S. B., Kammrath, A., Boyle, E. S., Kaser, L., Schnitzhofer, R., Graus, M., Turnipseed, A., Park, J. H., Weber, R. J., Hornbrook, R. S., Cantrell, C. A., Maudlin Iii, R. L., Kim, S., Nakashima, Y., Wolfe, G. M., Kajii, Y., Apel, E. C., Goldstein, A. H., Guenther, A., Karl, T., Hansel, A., and Keutsch, F. N.: Observations of glyoxal and formaldehyde as metrics for the anthropogenic impact on rural photochemistry, Atmos. Chem. Phys., 12, 9529-9543, https://doi.org/10.5194/acp-12-9529-2012, 2012.

Fan, C., de Leeuw, G., Yan, X., Dong, J., Kang, H., Fang, C., Li, Z., and Zhang, Y.: Evolution of aerosol optical depth over China in 2010–2024: increasing importance of meteorological influences, Atmos. Chem. Phys., 25, 11951-11973, https://doi.org/10.5194/acp-25-11951-2025, 2025.

Frieß, U., Beirle, S., Alvarado Bonilla, L., Bösch, T., Friedrich, M. M., Hendrick, F., Piters, A., Richter, A., van Roozendael, M., Rozanov, V. V., Spinei, E., Tirpitz, J. L., Vlemmix, T., Wagner, T., and Wang, Y.: Intercomparison of MAX-DOAS vertical profile retrieval algorithms: studies using synthetic data, Atmos. Meas. Tech., 12, 2155-2181, https://doi.org/10.5194/amt-12-2155-2019, 2019.

Hallgren, C., Körnich, H., Ivanell, S., and Sahlée, E.: A Single-Column Method to Identify Sea and Land Breezes in Mesoscale-Resolving NWP Models, Weather Forecasting, 38, 1025-1039, https://doi.org/10.1175/WAF-D-22-0163.1, 2023.

Hong, Q., Liu, C., Hu, Q., Zhang, Y., Xing, C., Ou, J., Tan, W., Liu, H., Huang, X., and Wu, Z.: Vertical distribution and temporal evolution of formaldehyde and glyoxal derived from MAX-DOAS observations: The indicative role of VOC sources, J Environ Sci (China), 122, 92-104, https://doi.org/10.1016/j.jes.2021.09.025, 2022.

Hong, Q., Xing, J., Xing, C., Yang, B., Su, W., Chen, Y., Zhang, C., Zhu, Y., and Liu, C.: Investigating vertical distributions and photochemical indications of formaldehyde, glyoxal, and $NO_2$ from MAX-DOAS observations in four typical cities of China, Sci. Total Environ., 954, https://doi.org/10.1016/j.scitotenv.2024.176447, 2024.

Hoque, H. M. S., Irie, H., and Damiani, A.: First MAX-DOAS Observations of Formaldehyde and Glyoxal in Phimai, Thailand, J. Geophys. Res.: Atmos., 123, 9957-9975, https://doi.org/10.1029/2018JD028480, 2018.

Jiang, Z., Wang, S., Yan, Y., Zhang, S., Xue, R., Gu, C., Zhu, J., Liu, J., and Zhou, B.: Constructing the 3D Spatial Distribution of the HCHO/NO$_2$ Ratio via Satellite Observation and Machine Learning Model, Environ. Sci. Technol., 59, 4047-4058, https://doi.org/10.1021/acs.est.4c12362, 2025.

Kaiser, J., Wolfe, G. M., Min, K. E., Brown, S. S., Miller, C. C., Jacob, D. J., deGouw, J. A., Graus, M., Hanisco, T. F., Holloway, J., Peischl, J., Pollack, I. B., Ryerson, T. B., Warneke, C., Washenfelder, R. A., and Keutsch, F. N.: Reassessing the ratio of glyoxal to formaldehyde as an indicator of hydrocarbon precursor speciation, Atmos. Chem. Phys., 15, 7571-7583, https://doi.org/10.5194/acp-15-7571-2015, 2015.

Kumar, V., Beirle, S., Dörner, S., Mishra, A. K., Donner, S., Wang, Y., Sinha, V., and Wagner, T.: Long-term MAX-DOAS measurements of NO$_2$, HCHO, and aerosols and evaluation of corresponding satellite data products over Mohali in the Indo-Gangetic Plain, Atmos. Chem. Phys., 20, 14183-14235, https://doi.org/10.5194/acp-20-14183-2020, 2020.

Lerot, C., Hendrick, F., Van Roozendael, M., Alvarado, L. M. A., Richter, A., De Smedt, I., Theys, N., Vlietinck, J., Yu, H., Van Gent, J., Stavrakou, T., Müller, J. F., Valks, P., Loyola, D., Irie, H., Kumar, V., Wagner, T., Schreier, S. F., Sinha, V., Wang, T., Wang, P., and Retscher, C.: Glyoxal tropospheric column retrievals from TROPOMI – multi-satellite intercomparison and ground-based validation, Atmos. Meas. Tech., 14, 7775-7807, https://doi.org/10.5194/amt-14-7775-2021, 2021.

Liang, Z. and Wang, D.: Sea breeze and precipitation over Hainan Island, Q. J. R. Meteorolog. Soc., 143, 137-151, https://doi.org/10.1002/qj.2952, 2017.

Liang, Z., Wang, D., Liu, Y., and Cai, Q.: A numerical study of the convection triggering and propagation associated with sea breeze circulation over Hainan Island, J. Geophys. Res.: Atmos., 122, 8567-8592, https://doi.org/10.1002/2016JD025915, 2017.

Liu, J., Li, X., Tan, Z., Wang, W., Yang, Y., Zhu, Y., Yang, S., Song, M., Chen, S., Wang, H., Lu, K., Zeng, L., and Zhang, Y.: Assessing the Ratios of Formaldehyde and Glyoxal to NO$_2$ as Indicators of O$_3$–NO$_x$–VOC Sensitivity, Environ. Sci. Technol., 55, 10935-10945, https://doi.org/10.1021/acs.est.0c07506, 2021.

MacDonald, S. M., Oetjen, H., Mahajan, A. S., Whalley, L. K., Edwards, P. M., Heard, D. E., Jones, C. E., and Plane, J. M. C.: DOAS measurements of formaldehyde and glyoxal above a south-east Asian tropical rainforest, Atmos. Chem. Phys., 12, 5949-5962, https://doi.org/10.5194/acp-12-5949-2012, 2012.

Martin, R. V., Parrish, D. D., Ryerson, T. B., Nicks Jr., D. K., Chance, K., Kurosu, T. P., Jacob, D. J., Sturges, E. D., Fried, A., and Wert, B. P.: Evaluation of GOME satellite measurements of tropospheric NO$_2$ and HCHO using regional data from aircraft campaigns in the southeastern United States, J. Geophys. Res.: Atmos., 109, https://doi.org/10.1029/2004JD004869, 2004.

Peng, S., Zheng, Y., Li, L., Gui, K., Zhu, J., Liu, S., Zhang, H., Zhao, H., Che, H., and Zhang, X.: Long-term variations in aerosol optical properties in Beijing: insights from diurnal and nocturnal continuous measurements, Atmos. Environ., 360, 121416, https://doi.org/10.1016/j.atmosenv.2025.121416, 2025.

Ren, B., Xie, P., Xu, J., Li, A., Tian, X., Hu, Z., Huang, Y., Li, X., Zhang, Q., Ren, H., and Ji, H.: Use of the PSCF method to analyze the variations of potential sources and transports of NO2, SO2, and HCHO observed by MAX-DOAS in Nanjing, China during 2019, Sci. Total Environ., 782, 146865, https://doi.org/10.1016/j.scitotenv.2021.146865, 2021.

Ren, H., Li, A., Xie, P., Hu, Z., Xu, J., Huang, Y., Li, X., Zhong, H., Zhang, H., Tian, X., Ren, B., Wang, S., Chai, W., and Du, C.: The Characterization of Haze and Dust Processes Using MAX-DOAS in Beijing, China, Remote Sens., 13, 5133, https://www.mdpi.com/2072-4292/13/24/5133, 2021.

Schreier, S. F., Richter, A., Peters, E., Ostendorf, M., Schmalwieser, A. W., Weihs, P., and Burrows, J. P.: Dual ground-based MAX-DOAS observations in Vienna, Austria: Evaluation of horizontal and temporal NO2, HCHO, and CHOCHO distributions and comparison with independent data sets, Atmos. Environ.: X, 5, 100059, https://doi.org/10.1016/j.aeaoa.2019.100059, 2020.

Schreier, S. F., Richter, A., Wittrock, F., and Burrows, J. P.: Estimates of free-tropospheric NO2 and HCHO mixing ratios derived from high-altitude mountain MAX-DOAS observations at midlatitudes and in the tropics, Atmos. Chem. Phys., 16, 2803-2817, https://doi.org/10.5194/acp-16-2803-2016, 2016.

Tan, W., Liu, C., Wang, S., Xing, C., Su, W., Zhang, C., Xia, C., Liu, H., Cai, Z., and Liu, J.: Tropospheric NO2, SO2, and HCHO over the East China Sea, using ship-based MAX-DOAS observations and comparison with OMI and OMPS satellite data, Atmos. Chem. Phys., 18, 15387-15402, https://doi.org/10.5194/acp-18-15387-2018, 2018.

Tang, X. L.: A Typical Sea-Land Breeze Process in Hainan Island, Applied Mechanics and Materials, 733, 387 - 390, 2015.

Vlemmix, T., Hendrick, F., Pinardi, G., De Smedt, I., Fayt, C., Hermans, C., Piters, A., Wang, P., Levelt, P., and Van Roozendael, M.: MAX-DOAS observations of aerosols, formaldehyde and nitrogen dioxide in the Beijing area: comparison of two profile retrieval approaches, Atmos. Meas. Tech., 8, 941-963, https://doi.org/10.5194/amt-8-941-2015, 2015.

Vrekoussis, M., Wittrock, F., Richter, A., and Burrows, J. P.: GOME-2 observations of oxygenated VOCs: what can we learn from the ratio glyoxal to formaldehyde on a global scale?, Atmos. Chem. Phys., 10, 10145-10160, https://doi.org/10.5194/acp-10-10145-2010, 2010.

Wagner, T., Beirle, S., Brauers, T., Deutschmann, T., Friess, U., Hak, C., Halla, J. D., Heue, K. P., Junkermann, W., Li, X., Platt, U., and Pundt-Gruber, I.: Inversion of tropospheric profiles of aerosol extinction and HCHO and $NO_2$ mixing ratios from MAX-DOAS observations in Milano during the summer of 2003 and comparison with independent data sets, Atmos. Meas. Tech., 4, 2685-2715, https://doi.org/10.5194/amt-4-2685-2011, 2011.

Wang, Y., Li, A., Xie, P.-H., Chen, H., Xu, J., Wu, F.-C., Liu, J.-G., and Liu, W.-Q.: Retrieving vertical profile of aerosol extinction by multi-axis differential optical absorption spectroscopy, Acta Physica Sinica, 62, 180705-180705, https://doi.org/10.7498/aps.62.180705, 2013.

Xia, G., Draxl, C., Optis, M., and Redfern, S.: Detecting and characterizing simulated sea breezes over the US northeastern coast with implications for offshore wind energy, Wind Energ. Sci., 7, 815-829, https://doi.org/10.5194/wes-7-815-2022, 2022.

Xing, C., Liu, C., Hong, Q., Liu, H., Wu, H., Lin, J., Song, Y., Chen, Y., Liu, T., Hu, Q., Tan, W., and Lin,

H.: Vertical distributions and potential sources of wintertime atmospheric pollutants and the corresponding ozone production on the coast of Bohai Sea, J. Environ. Manage., 319, 115721, https://doi.org/10.1016/j.jenvman.2022.115721, 2022.

Xing, C., Liu, C., Hu, Q., Fu, Q., Lin, H., Wang, S., Su, W., Wang, W., Javed, Z., and Liu, J.: Identifying the wintertime sources of volatile organic compounds (VOCs) from MAX-DOAS measured formaldehyde and glyoxal in Chongqing, southwest China, Sci. Total Environ., 715, 136258, https://doi.org/10.1016/j.scitotenv.2019.136258, 2020.

Zhang, R., Wang, S., Zhang, S., Xue, R., Zhu, J., and Zhou, B.: MAX-DOAS observation in the midlatitude marine boundary layer: Influences of typhoon forced air mass, J. Environ. Sci., 120, 63-73, https://doi.org/10.1016/j.jes.2021.12.010, 2022.

Zhang, S., Wang, S., Zhang, R., Guo, Y., Yan, Y., Ding, Z., and Zhou, B.: Investigating the Sources of Formaldehyde and Corresponding Photochemical Indications at a Suburb Site in Shanghai From MAX-DOAS Measurements, J. Geophys. Res.: Atmos., 126, https://doi.org/10.1029/2020jd033351, 2021.

Zhang, Y., Man, X., Zhang, S., Liu, L., Kong, F., Feng, T., and Liu, R.: Ground-based MAX-DOAS observations of formaldehyde and glyoxal in Xishuangbanna, China, J. Environ. Sci., 152, 328-339, https://doi.org/10.1016/j.jes.2024.04.036, 2025.

Zhang, Z.-z., Cao, C.-x., Song, Y., Kang, L., and Cai, X.-h.: STATISTICAL CHARACTERISTICS AND NUMERICAL SIMULATION OF SEA-LAND BREEZES IN HAINAN ISLAND, J. Trop. Meteorol., 20, 267-278, https://jtm.itmm.org.cn/en/article/id/20140309, 2014.

Zhao, D., Xin, J., Wang, W., Jia, D., Wang, Z., Xiao, H., Liu, C., Zhou, J., Tong, L., Ma, Y., Wen, T., Wu, F., and Wang, L.: Effects of the sea-land breeze on coastal ozone pollution in the Yangtze River Delta, China, Sci. Total Environ., 807, https://doi.org/10.1016/j.scitotenv.2021.150306, 2022.

---

## Author Comment (AC3)

**Response to Comments from Reviewer #1**

We thank you for the constructive comments and suggestions, which are very positive to improve scientific contents of the manuscript. We have revised the manuscript appropriately and addressed your comments point-by-point for consideration as below. The remarks from yours are shown in black, our responses (in blue) and the corresponding edits in the manuscript (in red) are shown below. All the page and line numbers mentioned following are refer to the revised manuscript without change tracked.

Reviewer #1: This manuscript presents an analysis of air pollutant daytime measurements of $NO_2$, HCHO and CHOCHO obtained using the MAX-DOAS technique in a rural coastal site in Hainan Island, China from June to August 2024. The analysis focuses on the dependence of air pollutant levels to the dominant airflow patterns at the measurement site, and distinguishes between days with and without sea-breeze, as well as days with tropical cyclones. The main results are: (i) days without sea breeze show higher pollutant concentrations compared to sea breeze days; (ii) typhoon days show vertical transport of pollutants to higher altitude; (iii) high values of glyoxal-to-$NO_2$ and HCHO-to-$NO_2$ ratios are indicative of a VOC-limited regime, and the glyoxal-to-$NO_2$ ratio is reliable for studying the ozone formation sensitivity. The article is generally well-written and the performed analysis seems to be robust, although I could not check the details because the MAX-DOAS dataset is not openly accessible. The findings of the paper concern pollutant patterns in coastal environments, but I am not convinced that they can be generalized to other environments. The article could be considered for publication once the following points are adequately addressed in a revised version.

R: We thank you for the constructive and encouraging assessment of our manuscript and for recognizing the value of our analysis. In accordance with the journal's data policy, we will make the MAX-DOAS dataset openly available upon publication to ensure transparency and facilitate independent verification of our results. Regarding the generalizability of our findings, we acknowledge that certain quantitative metrics are specific to Hainan's local conditions; however, the underlying physical and chemical processes identified in this study—such as the SB–driven cooling and suppression of vertical pollutant transport, pollutant redistribution under tropical cyclones, and the diagnostic value of photochemical indicators across different ACPs—are governed by universal mechanisms that also operate in many coastal and island environments. We have clarified these points and explicitly discussed both the limitations and the applicability of our conclusions in the revised manuscript.

Specific comments:

1. 1.158-167: Please mention that the ERA5-L fields are provided at 0.1 degree. Despite the finer resolution of ERA5-L compared to the parent ERA5 dataset, it is still relatively

coarse. An evaluation of the ERA5 or ERA5-L reanalysis against local meteorological measurements at the site should be included. It would be useful to evaluate ERA5-L under cyclonic conditions.

R: Thank you for your comment. We conducted a comparative analysis of temperature and relative humidity (RH) between ERA5 and the nearest meteorological station to our study site—Danzhou station (19.31°N, 109.35°E, 170 m a.s.l.) (Fig. R1). However, it should be noted that although Danzhou station is the closest meteorological site available, it still far away from the measurement site (~33 km). Such distance may also contribute to the discrepancies in temperature and humidity. Overall, the comparison shows relatively high agreement of temperature and RH between these two data sources.

[Figure]

*Figure R1. Correlation between Temperature (T_ERA5 and T_station) and Humidity (RH_ERA5 and RH_station) from ERA5_land and Danzhou Meteorological Station (19.31°N, 109.35°E) during summer 2024. The color of each hexagonal bin represents the frequency of data points within that grid cell. The yellow dashed line indicates the 1:1 line, and the green line shows the regression line between the two datasets.*

As shown in Fig. R2, the Danzhou station and ERA5_Land exhibit broadly consistent variations in WS and wind direction WD throughout the study period. The ERA5_Land WS is derived from the 10 m u- and v-wind components and represents grid-averaged conditions, whereas the Danzhou station is influenced by strong local surface roughness caused by nearby buildings and vegetation, resulting in lower observed WS. Despite these differences in magnitude, both datasets display similar prevailing wind directions on non-typhoon days, a shift toward southwesterly winds during typhoon periods, and comparable diurnal patterns in WS, as shown in Fig. R2(b), (c) and (d).

[Figure]

*Figure R2. Comparison of WS and WD between the Danzhou station and ERA5_Land. (a) Hourly variations in WS for the Danzhou station and ERA5_Land over the entire observation period; WD patterns at the (b) Danzhou station and (c) ERA5_Land during typhoon and non-typhoon days; (d) Diurnal variations of mean WS and WD for the Danzhou station and ERA5_Land on typhoon days. The gray shaded area indicates the range of the typhoon's duration. Wind direction is plotted in polar coordinates with percentage frequency indicated by concentric circles.*

Although ground-based observations can better capture near-surface meteorological conditions, their strong short-term fluctuations—particularly in wind fields—introduce substantial noise and complicate the identification of sea–land breeze circulations. More importantly, the Danzhou station is located far from the measurement site, making its data unrepresentative. In contrast, ERA5_Land provides smoother and more continuous spatiotemporal fields that coherently depict the evolution of large- and mesoscale wind structures. As a result, ERA5_Land has been widely applied in sea–land breeze research worldwide and is generally regarded as reliable and internally consistent for such analyses (Azorin-Molina et al., 2011; Hallgren et al., 2023; Xia et al., 2022; Zhao et al., 2022).

In summary, despite certain uncertainties, ERA5_Land offers physically consistent estimates of all required variables and supplies complete, continuous, and quality-controlled time series—advantages that ground-based observations cannot fully match. The comparison with station measurements further confirms good agreement between these two datasets. These features make ERA5_Land particularly suitable for investigating sea–land breeze circulations in this study.

2. l.208-210: The existence of additional sources is suggested here to explain the delayed HCHO enhancement (at 10am). Can you give more details? Note that the delayed HCHO peak formation could be due to VOCs that due to their longer lifetimes produce HCHO with delay. Please clarify.

R: Thank you for your comment. It should be clarified that our original statement, "linked to morning traffic emissions," was intended to emphasize this specific photochemical pathway: anthropogenic VOCs emitted by early morning traffic are oxidized by OH radicals under increasing solar radiation, leading to a delayed HCHO

peak around 10:00 local time. We acknowledge that our initial wording was somewhat ambiguous and did not fully convey the meaning. We have therefore revised the manuscript and please refer to Line 243-244.

In the manuscript:

"In contrast, the delayed HCHO peak (10:00 LT, VMRs = 5.14 ppbv) can be attributed to the photochemical oxidation of early morning traffic emitted VOCs under increasing solar radiation."

3. l.215-218: The residence time of CHOCHO and HCHO are actually quite similar, globally 2.9 h for CHOCHO and 5 h for HCHO. The profile shapes of Fig.3 for both species are also very similar, which contradicts the view expressed in the paper that CHOCHO remains confined within 500 m of the surface because of its shorter lifetime. In addition, it should be noted that methane oxidation contributes to HCHO at the higher levels (especially above 2km), where the CHOCHO levels are close to zero. Can you further elaborate why the lower photolysis rates would contribute to the late afternoon rebound for both species? I wonder why the reduced mixing, as illustrated in Fig.3, is not enough to explain the rebound. After all, reduced solar radiation simultaneously increases the lifetime and depletes the photochemical production of secondary species such as HCHO and CHOCHO. I recommend to include a figure similar to Fig.3 but showing the diurnal variation of the columns. Vertical mixing variations is expected to affect less the columns than the surface concentrations.

R: Thank you for your comment. We reconsider the comparison of HCHO and CHOCHO vertical profiles. In short, HCHO benefits from sustained and spatially widespread secondary production, such as methane oxidation and early-emitted VOC oxidation (Fortems-Cheiney et al., 2012; Liao et al., 2025; Wolfe et al., 2019). By contrast, CHOCHO is formed primarily through oxidation of later-emitted specific VOCs (e.g., aromatics and isoprene) (Chan Miller et al., 2017; Fu et al., 2008; Myriokefalitakis et al., 2008). Additionally, the much larger Henry's law constant of CHOCHO leads to more efficient uptake by humid aerosols, which further limits its upward transport (Du et al., 2025; Kroll et al., 2005; Sander, 2015; Waxman et al., 2015). Together, these factors offer a more plausible explanation for the different daytime vertical distributions of the two species. From the diurnal variation, it's easy to find that HCHO shows considerable abundance up to 800~1000 m while CHOCHO is more confined to lower altitudes, as shown in the left panels of Fig. 4 (b) and (c). We have revised the manuscript accordingly to clarify these points and remove the earlier ambiguity. Please refer to Line 249-252.

"In contrast, CHOCHO remains confined to lower altitudes (<500 m) (Fig. 4d, left), mainly because its weaker secondary formation at higher altitudes compared to HCHO and stronger uptake by humid aerosols that depletes its abundance (Fortems-Cheiney et al., 2012; Liao et al., 2025; Sander, 2015; Waxman et al., 2015; Wolfe et al., 2019)."

We acknowledge that explaining the late-afternoon rebound of HCHO and CHOCHO solely by reduced photolysis is insufficient, as weaker solar radiation also suppresses

their photochemical production, complicating the determination of source and sink contributions. According to your suggestion, we have added Fig. R3 (Fig. S9) to show the diurnal variations of surface AEC, $NO_2$, HCHO, and CHOCHO VMRs and column-integrated AOD, VCDs of $NO_2$, HCHO, and CHOCHO. The results indicate that both surface and column abundances of these species increase in the late afternoon, inferring that reduced mixing can not explain the rebound of these species enough. Meanwhile, the weakened photo-induced sink and enhanced anthropogenic emissions like traffic and residential cooking may increase the surface concentration and also contribute to the total column abundance. Nonetheless, reduced atmospheric mixing remains an important factor, allowing pollutants to accumulate near the surface. We have accordingly expanded the discussion in the main text and please refer to Line 252-254.

"The late afternoon rebound in HCHO and CHOCHO concentrations may be due to the combing effects of increased anthropogenic emissions of traffic and cooking activities, as well as reduced atmospheric mixing with temperatures drop and solar radiation decreases (Fig. S9)."

[Figure]

*Figure R3. Diurnal cycles of (a) surface AEC and AOD, along with surface VMRs and VCDs of (b) $NO_2$, (c) HCHO and (d) CHOCHO.*

4. The surface VMRs for HCHO and CHOCHO are different in Section 4 and in Section 3.1. Which one is correct?

R: Thank you for your comment. We would like to clarify that the surface VMRs of HCHO and CHOCHO reported in Section 3.1 refer to observational values from previous studies conducted in Beijing and Guangzhou. The values presented in Section 4 correspond to the actual observations obtained in this work. We have revised the manuscript accordingly to make this clearer. Please refer to Line 208-210.

"Notably, HCHO and CHOCHO concentrations ($4.33 \pm 1.07$ ppbv and $0.10 \pm 0.02$ ppbv, respectively) are comparable to values reported for summertime metropolitan regions, such as Beijing (HCHO: 4.41 ppbv) and Guangzhou (CHOCHO: 0.13 ppbv) (Hong et

al., 2024)."

5. The manuscript attributes the high glyoxal-to-$NO_2$ and HCHO-to-$NO_2$ ratios to BVOC sources and suggests that they are dominant compared to anthropogenic sources. Could you include bottom-up emission maps of BVOC, anthropogenic $NO_x$ and VOCs over the island?

R: Thank you for your comment. We have included Fig. S16 (Fig. R4) in the Supplement materials to show the bottom-up emissions of isoprene, anthropogenic $NO_x$, and VOCs over the island during summer 2024. The data are obtained from the CAMS with a spatial resolution of $0.25° \times 0.25°$ for isoprene and $0.1° \times 0.1°$ for anthropogenic $NO_x$ and VOCs (Copernicus Atmosphere Monitoring Service) global emission inventories (https://ads.atmosphere.copernicus.eu/datasets/cams-global-emission-inventories?tab=overview). It should be noted that CAMS currently provides isoprene as the only BVOC species for 2024, while other BVOC components are not yet available. Anthropogenic $NO_x$ and VOCs represent the total emissions from 12 major source sectors.

As shown in Figure R4, the isoprene emission intensity at the observation site is lower than that in densely vegetated areas such as Wuzhi Mountain. However, both anthropogenic $NO_x$ and VOCs emissions in this region are much lower than those in urban areas (e.g., Haikou and Sanya) and are also lower than the local isoprene flux. This emission pattern explains the relatively high glyoxal-to-$NO_2$ and HCHO-to-$NO_2$ ratios, especially considering that other BVOC sources (e.g., monoterpenes and sesquiterpenes) are not included. In addition, the spatial distribution clearly indicates that the upwind regions influenced by sea-breeze and typhoon flows are characterized by higher isoprene emissions and relatively low anthropogenic VOCs sources, supporting our conclusion that BVOC dominate in this area. The corresponding emission maps and related discussion have been added to the revised manuscript. Please refer to Line 193-195 and Line 415-419.

"Monthly emission fields of isoprene ($0.25° \times 0.25°$), anthropogenic $NO_x$, and VOCs ($0.1° \times 0.1°$) were also obtained from the CAMS (Copernicus Atmosphere Monitoring Service) global emission inventories (https://ads.atmosphere.copernicus.eu/datasets/cams-global-emission-inventories?tab=overview)."

"As shown in Fig. 9d, all ACPs demonstrated mean $R_{GF}$ values < 0.04 across altitudes, indicating BVOC dominance in the FK, which can be supported by the bottom-up emission maps from CAMS reanalysis data (Fig. S16). It shows higher emission than anthropogenic $NO_x$ and VOCs emissions—especially considering that other BVOC species were not included here, e.g., monoterpenes and sesquiterpenes."

[Figure]

*Figure R4. Monthly mean of (a–c) isoprene, (d–f) anthropogenic VOCs (AVOC), and (g–i) anthropogenic NOₓ emission fluxes (EF) over Hainan Island for June–August 2024.*

6. The manuscript should include a comparison with air pollutant levels from other coastal locations or islands based on literature studies. Besides the typhoon days, what is special about Hainan Island? Can we generalize the findings to other locations?

R: Thank you for your comment. We have appropriately supplemented the manuscript with a comparative analysis of air pollutant concentrations between this study and other coastal regions or islands. Please refer to Line 201-214.

"The observed summer aerosol loading (mean AEC and AOD measuring $0.11 \pm 0.03$ km-1 and $0.29 \pm 0.1$, respectively) exhibited markedly lower values compared to urban agglomerations like Beijing and Shanghai (where AOD is around 0.4) (Fan et al., 2025; Peng et al., 2025), and slightly higher than in coastal Thailand, where monsoonal rainfall efficiently scavenges aerosols (Peengam et al., 2025). The average $NO_2$ surface VMRs ($1.61 \pm 0.53$ ppbv) was 3 to 5 times lower than the reported values in major Chinese megacity centers, such as the Beijing–Tianjin–Hebei region ($7.62 \pm 1.39$ ppbv) and the Yangtze River Delta ($7.45 \pm 0.87$ ppbv) (Lou et al., 2025; Ministry of Ecology and Environment of the People's Republic of China, 2024a, b, c), reflecting minimal local traffic and industrial emissions. Notably, HCHO and CHOCHO concentrations ($4.33 \pm 1.07$ ppbv and $0.10 \pm 0.02$ ppbv, respectively) are comparable to values reported for summertime metropolitan regions, such as Beijing (HCHO: 4.41 ppbv) and Guangzhou (CHOCHO: 0.13 ppbv) (Hong et al., 2024). This likely stems from biogenic emissions from surrounding tropical ecosystems compensating for diminished anthropogenic sources. However, the observed HCHO and CHOCHO substantially

higher than values reported in less anthropogenically influenced islands and coastal regions, including the coastal area of Mt. Lao, Qingdao, the Galápagos Islands, and Cape Verde (Lawson et al., 2015; Mahajan et al., 2014; Mahajan et al., 2010; Zhao et al., 2024)."

Similar to the Community Comment #14, we have added a dedicated paragraph in the conclusion section outlining the extent to which our findings can be generalized and the key limitations of our study. Please refer to Line 535-543.

"This study reveals several summertime physical and chemical processes that are characteristic of Hainan Island, so the quantitative values reported here—such as pollutant concentrations, ORA parameter ranges and FNR/GNR thresholds—are shaped by the island's unique land–sea configuration, monsoon regime, and emission environment and should be treated as site-specific. However, the SB–driven cooling and suppression of vertical pollutant transport, the typhoon-related scavenging, redistribution, and uplift, and the diagnostic value of different photochemical indicators under different ACPs are processes that commonly occur across many coastal and island settings. Thus, while specific numerical thresholds may vary, the underlying conceptual processes and the ORA framework offer a transferable basis for interpreting similar coastal phenomena, particularly in regions with comparable climatic and geographic conditions."

Technical comments:

7. l.40: read 'due to the fact that it originates...'

R: Thank you for your comment. We have revised the sentence and please refer to Line 49:

"$NO_2$ concentrations typically decrease exponentially with altitude due to the fact that it originates mainly from surface-level anthropogenic emissions, …"

8. l.47: remove 'regimes' (repetition)

R: Thank you for your comment. We have removed the repeated word. Please refer to Line 56-57.

9. l.52: 'Coastal atmospheric environments are significantly influenced by sea-land breeze circulation and typhoons': Typhoons only affect a specific region of the globe. Rephrase.

R: Thank you for your comment. We have revised the sentence to clarify that. Please refer to Line 62-63.

"Coastal atmospheric environments are significantly influenced by sea-land breeze (SLB) circulation and also typhoons in some regions (e.g., the western North Pacific)."

10. l.63: 'in island', something is missing here

R: Thank you for your comment. We have revised the sentence to correct the grammatical structure. Please refer to Line 72-75.

"Nevertheless, research gaps persist regarding vertical distributions of $NO_2$, HCHO, and CHOCHO under SB and typhoon conditions, particularly in island where complex topography and unique atmospheric environments may cause these pollutants to behave differently than over continental or coastal regions."

11. l.64: 'located far from mainland China', be more specific

R: Thank you for your comment. We have updated the description and please refer to Line 76.

"Hainan Island is located away from China's mainland and lies approximately 20 km from the nearest point in Guangdong Province."

12. l.64-72: Add information about the island (area, population, large cities, powerplants, fraction of vegetation)

R: Thank you for your comment. As suggested, we have integrated key information about the island. Please refer to Line 77-81.

"The island spans approximately 32,900 $km^2$ and supports a population of 10.48 million people concentrated in coastal cities (e.g., Haikou, Sanya, and Danzhou), while also maintaining dense vegetation with over 60% forest coverage featuring unique tropical ecosystems (Hainan Provincial Bureau of Statistics, 2024). Adjacent to the South China Sea, its topography and geographic position render the island frequently influenced by SB and typhoons (Fu et al., 2023; Liang and Wang, 2017; Zhang et al., 2014)."

13. l.66: 'Given the superior air quality', sounds weird. Replace 'superior' by 'good'

R: Thank you for your comment. We have replaced "superior" with "good" in the revised manuscript and please refer to Line 81.

14. l.72: 'are still unclear'. Replace by 'are not investigated yet'

R: Thank you for your comment. We have revised the text accordingly, and please refer to Line 86-87.

15. l.73: read 'measurements'

Response: Thank you for your comment. We have corrected the text to "measurements." Please refer to Line 88.

16. l.83: Beibu Gulf is not shown on Fig.1

Response: Thank you for pointing this out. Since "Beibu Gulf" is the Chinese name for the region, while "Gulf of Tonkin" (in Fig. 1) is the internationally recognized name for this region. To ensure consistency and avoid confusion, we have revised the manuscript to use the internationally recognized name, "Gulf of Tonkin", throughout both the text and figures. Please refer to Line 97-98.

"The observation site (altitude approx. 100 m above sea level) lies within a topographically complex region bordered to the northwest by the Gulf of Tonkin and to the southeast by the Wuzhi Mountain (Fig. 1a)."

17. l:83: 'Wuzhi mountain (18.9°N, 109.7°E)'. What are these coordinates?

R: Thank you for your comment. We have removed the coordinates, as they specifically refer to the Wuzhi Mountain main peak and are not necessary for the context. Please refer to Line 98.

18. l.91: add an 'r' to 'dime'

R: Thank you for pointing this out. We have corrected the typo, so it now reads "dimer". Please refer to Line 107.

19. l.97: remove 's' from 'Supplements', here and elsewhere in the paper

R: Thank you for your comment. We have removed the extra "s" and revised all instances throughout the manuscript and please refer to Line 113, 171 and 354.

20. Fig.1a: The color of highways is orange, not yellow

R: Thank you for your comment. We have revised the figure caption to correctly describe the highway color as orange instead of yellow. Please refer to Line 137.

21. l.146-156: Fig.S4 is central to the analysis and the filters mentioned are not explained in the main manuscript. Either include a new section on ORA and move Fig.S4 to the main manuscript, or keep it as is and include a brief description on the filters in the main manuscript. Could you also provide the percentage of the days identified as sea breeze days or refer to the relevant section?

R: Thank you for your comment. Following your suggestion, we have moved Fig. S4 to the manuscript as Fig. 2 now shown in Section 2.2. To keep the fluency and readability of the manuscript, we have kept the detailed parameter definitions and justifications in the Supplementary Text S1, while the main manuscript now includes a

concise description of the purpose and function of each filtering step for clarity. In addition, the specific dates identified as sea-breeze days are provided in Table S3 of the Supplement, and the percentage of such days is discussed in Section 3.2. Please refer to Line 170-171.

"The specific functions of the modules and filters are detailed in Text S2 (Supplement) and identification results are summarized in Table S3 and Section 3.2."

22. l.170: 'series daily', add 'of' between the two

R: Thank you for your comment. We have corrected the phrase to be "series of daily." Please refer to Line 198.

23. l.179: 'This likely steems from biogenic emissions'. Could you provide biogenic emission maps?

R: Thank you for your comment. We have added emission maps of isoprene, anthropogenic $NO_x$, and VOCs in the Supplementary Material (Fig. S16) and incorporated corresponding discussions in the revised manuscript. Please also refer to response to Comment #5 for more details above.

24. l.181: 'refined', can you be more specific?

R: Thank you for your comment. We have revised the manuscript to specify the temporal resolution more clearly and please refer to Line 215.

"At an hourly temporal resolution (Fig. S6), HCHO and CHOCHO exhibited…. "

25. l.206: Replace 'The early peaks' by 'The high HCHO and CHOCHO morning levels'

R: Thank you for your comment. We have revised the text accordingly, and please refer to Line 241-242.

26. l.207: 'in the previous night', replace by 'during daytime'

R: Thank you for your comment. We have revised the text, replacing "in the previous night" with "during daytime" as suggested. Please refer to Line 242.

27. l.209: remove 'beyond this mechanism'

Response: Thank you for your comment. We have removed it as suggested. Please refer to Line 244.

28. l.212: replace 'A' by 'a'

Response: Thank you for your comment. We have corrected the capitalization, and please refer to Line 247.

29. Section 3.3. Move Table S5 to the main manuscript to quick reference

Response: Thank you for your comment. In accordance with your suggestion, we have moved Table S5 to Section 3.3 as Table 1 in the main manuscript.

30. l. 326: Yang et al. 2024. Replace by a textbook reference here

Response: Thank you for your comment. We have updated the citation and please refer to Line 367.

31. l.326: add space '1and'

Response: Thank you for your comment. We have added the missing space. Please refer to Line 367.

32. l.379: 'to the local area'. Do you mean the measurement site?

Response: Thank you for your comment. Yes, we mean the measurement site. We have revised the text to "to the measurement site" for clarity. Please refer to Line 423.

33. l.456: '...several megacities', Please provide reference here

Response: Thank you for your comment. We have added an appropriate reference to support the statement regarding several megacities and please refer to Line 502.

**Reference:**

Azorin-Molina, C., Tijm, S., and Chen, D.: Development of selection algorithms and databases for sea breeze studies, Theor. Appl. Climatol., 106, 531-546, https://doi.org/10.1007/s00704-011-0454-4, 2011.

Chan Miller, C., Jacob, D. J., Marais, E. A., Yu, K., Travis, K. R., Kim, P. S., Fisher, J. A., Zhu, L., Wolfe, G. M., Hanisco, T. F., Keutsch, F. N., Kaiser, J., Min, K. E., Brown, S. S., Washenfelder, R. A., González Abad, G., and Chance, K.: Glyoxal yield from isoprene oxidation and relation to formaldehyde: chemical mechanism, constraints from SENEX aircraft observations, and interpretation of OMI satellite data, Atmos. Chem. Phys., 17, 8725-8738, https://doi.org/10.5194/acp-17-8725-2017, 2017.

Du, C., Wang, H., Gao, Y., Yan, R., Jing, S. a., Zhou, M., Wang, Q., Lou, S., Huang, C., Huang, D. D., Shang, Y., and An, J.: Processes Driving Diurnal and Seasonal Variations of Formaldehyde in an Urban Environment, ACS Earth Space Chem., 9, 1174-1184, https://doi.org/10.1021/acsearthspacechem.5c00020, 2025.

Fortems-Cheiney, A., Chevallier, F., Pison, I., Bousquet, P., Saunois, M., Szopa, S., Cressot, C., Kurosu, T. P., Chance, K., and Fried, A.: The formaldehyde budget as seen by a global-scale multi-constraint and multi-species inversion system, Atmos. Chem. Phys., 12, 6699-6721, https://doi.org/10.5194/acp-12-6699-2012, 2012.

Fu, C., Dan, L., Tong, J., and Xu, W.: Influence of Typhoon Nangka Process on Ozone Pollution in Hainan Island, Huanjing Kexue, 44, 2481-2491, https://doi.org/10.13227/j.hjkx.202206123, 2023.

Fu, T.-M., Jacob, D. J., Wittrock, F., Burrows, J. P., Vrekoussis, M., and Henze, D. K.: Global budgets of atmospheric glyoxal and methylglyoxal, and implications for formation of secondary organic aerosols, J. Geophys. Res.: Atmos., 113, https://doi.org/10.1029/2007JD009505, 2008.

Hainan Provincial Bureau of Statistics (Ed.) Hainan Statistical Yearbook 2024, China Statistics Press, Beijing, https://stats.hainan.gov.cn/tjj/tjsu/ndsj/2024/202412/P020250116308974141111.pdf, 2024.

Hallgren, C., Körnich, H., Ivanell, S., and Sahlée, E.: A Single-Column Method to Identify Sea and Land Breezes in Mesoscale-Resolving NWP Models, Weather Forecasting, 38, 1025-1039, https://doi.org/10.1175/WAF-D-22-0163.1, 2023.

Hong, Q., Xing, J., Xing, C., Yang, B., Su, W., Chen, Y., Zhang, C., Zhu, Y., and Liu, C.: Investigating vertical distributions and photochemical indications of formaldehyde, glyoxal, and $NO_2$ from MAX-DOAS observations in four typical cities of China, Sci. Total Environ., 954, https://doi.org/10.1016/j.scitotenv.2024.176447, 2024.

Kroll, J. H., Ng, N. L., Murphy, S. M., Varutbangkul, V., Flagan, R. C., and Seinfeld, J. H.: Chamber studies of secondary organic aerosol growth by reactive uptake of simple carbonyl compounds, J. Geophys. Res.: Atmos., 110, https://doi.org/10.1029/2005JD006004, 2005.

Liang, Z. and Wang, D.: Sea breeze and precipitation over Hainan Island, Q. J. R. Meteorolog. Soc., 143, 137-151, https://doi.org/10.1002/qj.2952, 2017.

Liao, J., Wolfe, G. M., Kotsakis, A. E., Nicely, J. M., St. Clair, J. M., Hanisco, T. F., González Abad, G., Nowlan, C. R., Ayazpour, Z., De Smedt, I., Apel, E. C., and Hornbrook, R. S.: Validation of formaldehyde products from three satellite retrievals (OMI SAO, OMPS-NPP SAO, and OMI BIRA) in the marine atmosphere with four seasons of Atmospheric Tomography Mission (ATom) aircraft observations, Atmos. Meas. Tech., 18, 1-16, https://doi.org/10.5194/amt-18-1-2025, 2025.

Myriokefalitakis, S., Vrekoussis, M., Tsigaridis, K., Wittrock, F., Richter, A., Brühl, C., Volkamer, R., Burrows, J. P., and Kanakidou, M.: The influence of natural and anthropogenic secondary sources on the glyoxal global distribution, Atmos. Chem. Phys., 8, 4965-4981, https://doi.org/10.5194/acp-8-4965-2008, 2008.

Sander, R.: Compilation of Henry's law constants (version 4.0) for water as solvent, Atmos. Chem. Phys., 15, 4399-4981, https://doi.org/10.5194/acp-15-4399-2015, 2015.

Waxman, E. M., Elm, J., Kurtén, T., Mikkelsen, K. V., Ziemann, P. J., and Volkamer, R.: Glyoxal and Methylglyoxal Setschenow Salting Constants in Sulfate, Nitrate, and Chloride Solutions: Measurements and Gibbs Energies, Environ. Sci. Technol., 49, 11500-11508, https://doi.org/10.1021/acs.est.5b02782, 2015.

Wolfe, G. M., Nicely, J. M., St. Clair, J. M., Hanisco, T. F., Liao, J., Oman, L. D., Brune, W. B., Miller, D., Thames, A., González Abad, G., Ryerson, T. B., Thompson, C. R., Peischl, J., McKain, K.,

Sweeney, C., Wennberg, P. O., Kim, M., Crounse, J. D., Hall, S. R., Ullmann, K., Diskin, G., Bui, P., Chang, C., and Dean-Day, J.: Mapping hydroxyl variability throughout the global remote troposphere via synthesis of airborne and satellite formaldehyde observations, Proceedings of the National Academy of Sciences, 116, 11171-11180, https://doi.org/10.1073/pnas.1821661116, 2019.

Xia, G., Draxl, C., Optis, M., and Redfern, S.: Detecting and characterizing simulated sea breezes over the US northeastern coast with implications for offshore wind energy, Wind Energ. Sci., 7, 815-829, https://doi.org/10.5194/wes-7-815-2022, 2022.

Zhang, Z.-z., Cao, C.-x., Song, Y., Kang, L., and Cai, X.-h.: STATISTICAL CHARACTERISTICS AND NUMERICAL SIMULATION OF SEA-LAND BREEZES IN HAINAN ISLAND, J. Trop. Meteorol., 20, 267-278, https://jtm.itmm.org.cn/en/article/id/20140309, 2014.

Zhao, D., Xin, J., Wang, W., Jia, D., Wang, Z., Xiao, H., Liu, C., Zhou, J., Tong, L., Ma, Y., Wen, T., Wu, F., and Wang, L.: Effects of the sea-land breeze on coastal ozone pollution in the Yangtze River Delta, China, Sci. Total Environ., 807, https://doi.org/10.1016/j.scitotenv.2021.150306, 2022.